# Contradiction Retrieval via Contrastive Learning with Sparsity

**Haike Xu** [* 1]   **Zongyu Lin** [* 2]   **Kai-Wei Chang** [2]   **Yizhou Sun** [2]   **Piotr Indyk** [1]

## Abstract

Contradiction retrieval refers to identifying and extracting documents that explicitly disagree with or refute the content of a query, which is important to many downstream applications like fact checking and data cleaning. To retrieve contradiction argument to the query from large document corpora, existing methods such as similarity search and cross-encoder models exhibit different limitations. To address these challenges, we introduce a novel approach: SparseCL that leverages specially trained sentence embeddings designed to preserve subtle, contradictory nuances between sentences. Our method utilizes a combined metric of cosine similarity and a sparsity function to efficiently identify and retrieve documents that contradict a given query. This approach dramatically enhances the speed of contradiction detection by reducing the need for exhaustive document comparisons to simple vector calculations. We conduct contradiction retrieval experiments on Arguana, MSMARCO, and HotpotQA, where our method produces an average improvement of $11.0\%$ across different models. We also validate our method on downstream tasks like natural language inference and cleaning corrupted corpora. This paper outlines a promising direction for non-similarity-based information retrieval which is currently underexplored.

## 1. Introduction

Training sentence embedding for similarity retrieval has been well studied in the literature (Gao et al., 2021; Xiong et al., 2020; Karpukhin et al., 2020), where a standard practice is to use contrastive learning to map those similar sentences together and those dissimilar sentences far from each other. However, these existing sentence embed-

dings are mainly tailored to similarity retrieval, while as far as we know, there hasn't been sentence embeddings for non-similarity based retrieval. In this paper, we study the problem of contradiction retrieval, a typical case of non-similarity based retrieval. Given a large document corpus and a query passage, the goal is to retrieve document(s) in the corpus that contradict the query, assuming they exist. This problem has a large number of applications, including counter-argument detection (Wachsmuth et al., 2018) and fact verification (Thorne et al., 2018). The standard approaches to retrieving contradictions are two-fold. One is to use a bi-encoder (Xiao et al., 2023; Li & Li, 2023; Li et al., 2023) that maps each document to a feature space such that two contradicting documents are mapped close to each other (e.g., according to the cosine metric) and use nearest neighbor search algorithms. The second approach is to train a cross-encoder model (Xiao et al., 2023) that determines whether two documents contradict each other, and apply it to each document or passage in the corpus.

Unfortunately, both methods suffer from limitations. The first approach (cosine similarity search on sentence embeddings) is inherently incapable of representing the "contradiction relation" between the documents, due to the fact that the cosine metric is "transitive" (See Appendix B for formal analysis): if $A$ is similar to $B$, and $B$ is similar to $C$, then $A$ is also similar to $C$. As an example, consider an original sentence and its paraphrase in Table 12. Both of them contradict the sentence in the third column but they are not contradicting each other. The second approach, which uses a cross-encoder model, can capture the contradiction between sentences to some extent, but it is much more computationally expensive. Our experiment in Appendix F shows that compared with standard vector computation, running a cross-encoder is at least 200 times slower.

In this paper, we propose to overcome these limitations by introducing SPARSECL for efficient contradiction retrieval using sparse-aware sentence embeddings. The key idea behind our approach is to train a sentence embedding model to preserve *sparsity of differences* between the contradicted sentence embeddings. When answering a query, we calculate a score between the query and each document in the corpus, based on *both* the cosine similarity and the sparsity of the difference between their embeddings, and retrieve the ones with the highest scores. Our specific measure of

---

[*]Equal contribution  [1]MIT  [2]University of California, Los Angeles. Correspondence to: Haike Xu <haikexu@mit.edu>.

*Proceedings of the 42$^{nd}$ International Conference on Machine Learning*, Vancouver, Canada. PMLR 267, 2025. Copyright 2025 by the author(s).

sparsity is defined by the Hoyer measure of sparsity (Hurley & Rickard, 2009), which uses the scaled ratio of the $\ell_1$ norm and the $\ell_2$ norm of a vector as a proxy of the number of non-zero entries in the vector. Unlike the cosine metric, the Hoyer measure is not transitive (please refer to Appendix B for a detailed analysis), which avoids the limitations of the former. At the same time this method is much more efficient than a cross-encoder, as both the cosine metric and the Hoyer measure are easy to compute given the embeddings. The Hoyer sparsity histogram of our trained embeddings is displayed in Figure 2.

We first evaluate our method on the counter-argument detection dataset Arguana (Wachsmuth et al., 2018), which to the best of our knowledge, is the only publicly available dataset suitable for testing contradiction retrieval. In addition, we generate another two data sets, where contradictions for documents in MSMARCO (Nguyen et al., 2016) and HotpotQA (Yang et al., 2018) datasets are generated using GPT-4 (Achiam et al., 2023). Our experiments demonstrate the efficacy of our approach in contradiction retrieval, as seen in Table 1. We also apply our method to corrupted corpus cleaning problem, where the goal is to filter out contradictory sentences in a corrupted corpus and preserve good QA retrieval accuracy.

To summarize. our contributions can be divided into three folds:

- We introduce a novel contradiction retrieval method that employs specially trained sentence embeddings combined with a metric that includes both cosine similarity and the Hoyer measure of sparsity. This approach effectively captures the essence of contradiction while being computationally efficient.

- Our method demonstrates superior contradiction retrieval metrics over different datasets compared to existing methods. This underscores the effectiveness of our embedding and scoring approach.

- We apply our contradiction retrieval method to two downstream settings: (1) corpus cleaning, where SPARSECL removes contradictions from corrupted datasets to maintain high-quality QA retrieval; and (2) natural language inference, where SPARSECL assigns a higher Hoyer sparsity score between contradicted pairs. These applications highlight the practical benefits of our approach in real-world scenarios.

## 2. Related Work

**Counter Argument Retrieval**    A direct application of our contradiction retrieval task in "counter-argument retrieval". Since the curation of Arguana dataset by (Wachsmuth et al.,

2018), there has been a few previous work on retrieving the best counter-argument for a given argument (Orbach et al., 2020; Shi et al., 2023). In terms of methods, (Wachsmuth et al., 2018) uses a weighted sum of different word and embedding similarities and (Shi et al., 2023) designs a "Bipolar-encoder" and a classification head. We believe that our method relying only on cosine similarity and sparsity is simpler than theirs and produces better results in the experiment. In addition, some analyses in the counter-argument retrieval papers are specific to the "debate" setting, e.g. they rely on topic, stance, premise/conclusion, and some other inherent structures in debates for help, which may prevent their methods from being generalized to broader scenarios.

**Fact verification and LLM hallucination**    Addressing the hallucination problem in Large Language Models has been a subject of many research efforts in recent years. According to the three types of different hallucinations in (Zhang et al., 2023b), here we only focus on those so called "Fact-Conflicting Hallucination" where the outputs of LLM contradict real world knowledge. The most straightforward way to mitigate this hallucination issue is to assume an external groundtruth knowledge source and augment LLM's outputs with an information retrieval system. There have been a few works on this line showing the success of this method (Ren et al., 2023; Mialon et al., 2023). This practice is very similar to "Fact-Verification" (Thorne et al., 2018; Schuster et al., 2021) where the task is to judge whether a claim is true or false based on a given knowledge base.

However, as pointed out by (Zhang et al., 2023b), in the era of LLM, the external knowledge base can encompass the whole internet. It is impossible to assume that all the information there are perfectly correct and there may exist conflicting information within the database. In the context of our paper, instead of using a groundtruth database to check an external claim, our goal is to check the internal contradictions between different documents in an unknown corpus.

**Learning augmented LLM and retrieval corpus attack** Augmenting large language models with retrieval has been shown to be useful for many purposes. Recently, there have been a few works (Zhong et al., 2023; Zou et al., 2024) studying the vulnerability of retrieval system from adversarial attack. Specifically, they show that adding a few corrupted data points to the corpus will significantly drop the retrieval accuracy. This phenomenon brings our attention to the necessity of checking the factuality of the knowledge database. Note that the type of corrupted documents considered by their papers are different from ours. While they consider the injection of adversarially generated documents, we consider the existence of contradicted documents as a natural part of the corpus. Also their purpose is to show the effect of

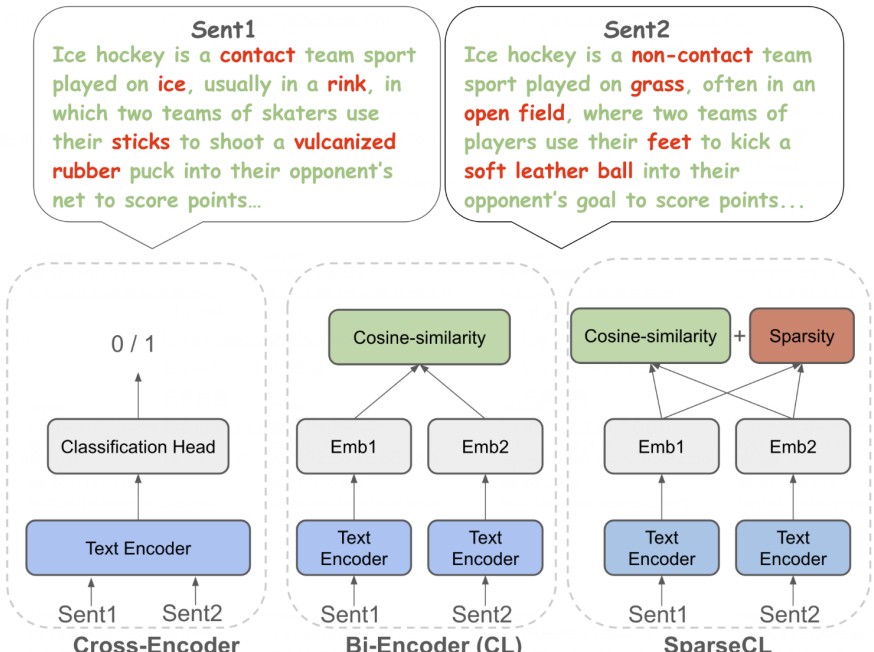

*Figure 1.* Comparison of our SPARSECL with Cross-Encoder and Contrastive-Learning based Bi-Encoder for contradiction retrieval.

adversarial attack, while we provide a defense method for a certain kind of corrupted database.

## 3. Method

**Problem Formulation**    We consider the contradiction retrieval problem: given a passage corpus $C = \{p_1, p_2, ...p_n\}$ and a query passage $q$, retrieve the "best" passage $p^*$ that contradicts $q$. We assume that several similar passages supporting $q$ might exist in the corpus $C$.

**Embedding based method**    Judging whether two passages contradict each other is a standard Natural Language Inference task and can be easily tackled by many off-the-shelf language models (Touvron et al., 2023; Xu et al., 2022). However, to retrieve the best candidate from the corpus, we have to iterate the whole corpus, or at least send the candidates retrieved by similarity search to the language model to determine if they constitute contradiction. This is time consuming, given that there are potentially many similar passages in the corpus. Therefore, in our paper, we mainly focus on those methods that only rely on their passage embeddings. Specifically, we want to design a simple scoring function $F$ that given the embeddings of two passages, outputs a score between 0 and 1, indicating the likelihood that they are contradicting each other.

**Sparse Aware Embeddings**    Following the idea from counter-argument retrieval papers (Wachsmuth et al., 2018), such a score function should be a combination of similarity and dissimilarity functions. Observe that a dissimilarity

function is basically a negation of a similarity function, so Wachsmuth et al. (2018) proposes several different similarity functions and sets the scoring function to maximize one of them and minimize another. Here, instead of enumerating different similarity functions, we consider another notion: the "sparsity" of their embedding differences. The basic intuition is as follows. Suppose that all sentences are represented as vectors in a "semantic" basis, where each coordinate represents one clearly identifiable semantic meaning. Then a contradiction between two passages should manifest itself as a difference in a few coordinates, while other coordinates should be quite close to each other. The issue, however, is that we do not know how to construct the appropriate basis, and the sparsity is defined with respect to a fixed coordinate system. Nevertheless, following this intuition, we fine-tune sentence embedding models using contrastive learning, by rewarding the sparsity of the difference vectors between embeddings of contradicting passages. Please see Figure 2 for the Hoyer sparsity histogram of our trained embeddings.

**SPARSECL**    We use contrastive learning (Gao et al., 2021; Karpukhin et al., 2020) to fine-tune a pretrained sentence embedding model to generate the desired sparsity-aware embeddings. The choice of positive and negative examples are exactly the reverse of the choice we make when the training sets are Natural Language Inference datasets. The positive example for a passage is its contradiction passage in the training set. The hard negative example for a passage is its similar passage in the training set. There are also other random in-batch passages as soft negative examples. The

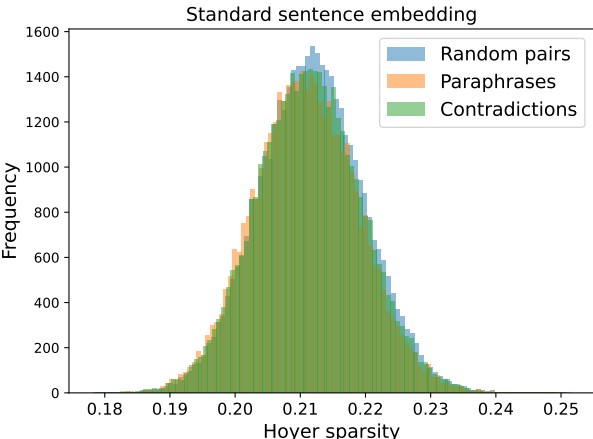

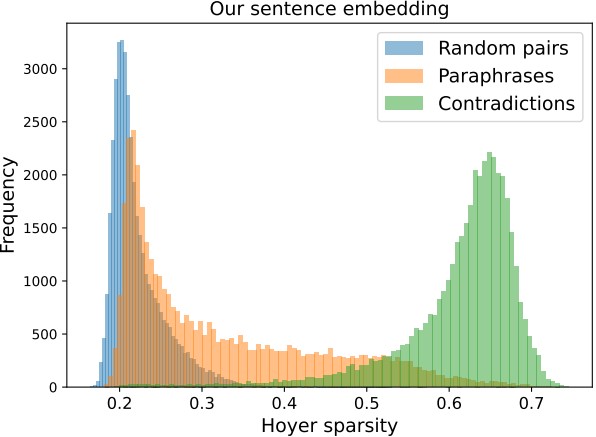

*Figure 2.* Histograms for the Hoyer sparsity of different pairs of sentence embedding differences on HotpotQA test set. The upper figure is the histogram produced by a standard sentence embedding model ("bge-base-en-v1.5"), where the median Hoyer sparsity values for random pairs, paraphrases, and contradictions are $0.212, 0.211, 0.211$. The lower figure is the histogram produced by our sentence embedding model fine-tuned from "bge-base-en-v1.5" using our SPARSECL method, where the median Hoyer sparsity values for random pairs, paraphrases, and contradictions are $0.212, 0.281, 0.632$.

sparsity function we choose here is Hoyer sparsity function from (Hurley & Rickard, 2009). Let $h_1$ and $h_2$ be two sentence embeddings and their embeddings have dimension $d$. We define

$$\text{Hoyer}(h_1, h_2) = \left( \sqrt{d} - \frac{\|h_1 - h_2\|_1}{\|h_1 - h_2\|_2} \right) \Big/ \left( \sqrt{d} - 1 \right).$$

This is a transformed version of the ratio of the $l_1$ to the $l_2$ norm, with output normalized to $[0, 1]$.

Finally, for each training tuple $(x_i, x_i^+, x_i^-)$ with their embeddings $(h_i, h_i^+, h_i^-)$, batch size $N$, and temperature $\tau$, its loss function is defined as

$$l_i = -\log \frac{e^{\text{Hoyer}(h_i, h_i^+)/\tau}}{\sum_{j=1}^{N} \left( e^{\text{Hoyer}(h_i, h_j^+)/\tau} + e^{\text{Hoyer}(h_i, h_j^-)/\tau} \right)}.$$

**Scoring function for contradiction retrieval**   For the score function for contradiction retrieval, we use a weighted sum of the standard cosine similarity and our sparsity function. Note that the cosine similarity is provided separately by any off-the-shelf sentence embedding model in a zeroshot manner. It can also be fine-tuned. Let $E()$ be the standard sentence embedding model and $E_s()$ be our sparse-aware sentence embedding model trained by SPARSECL. Then the final score function for contradiction retrieval is

$$F(q, p) = \cos\left(E(q), E(p)\right) + \alpha \cdot \text{Hoyer}(E_s(q), E_s(p)).$$

where $\alpha$ is a scalar tuned using the validation set. Note that the criterion for contradiction is usually case-dependent, so it is necessary that we reserve a parameter to adapt to different notions of contradiction. To get the answer passages, we calculate the score function for all passages and report the top 10 of them[1].

## 4. Experiments

We test our SPARSECL method on a counterargument retrieval task Arguana (Wachsmuth et al., 2018) and two contradiction retrieval datasets adapted from HotpotQA (Yang et al., 2018) and MSMARCO (Nguyen et al., 2016). Then, we apply our contradiction retrieval task to two downstream applications: retrieval corpus cleaning and natural language inference. Finally, we perform ablation studies to explain the functionality of each component of our method. Most of our experiments are not so computationally extensive, which can be run by one single A6000 GPU. We run our major experiments on A6000 and A100 GPUs.

### 4.1. Counter-argument Retrieval

**Dataset**   Arguana is a dataset curated in (Wachsmuth et al., 2018), where the author provide a corpus of 6753 argument-counterargument pairs, taken from 1069 debates with 15 themes on idebate.org. For each debate, the arguments are further divided into two opposing stances (pro and con). For each stance, there are paired arguments and counter-arguments. The dataset is split into the training set (60% of the data), the validation set (20%), and the test set (20%). This ensures that data from each individual debate is included in only one set and that debates from every theme are represented in every set. The task goal is: given an argument, retrieve its best counter-argument.

**Training**   We use Arguana's training set to fine-tune our sparsity aware sentence embedding model via SPARSECL.

---

[1]In the actual implementation, for time efficiency, we first use FAISS (Douze et al., 2024) to retrieve the top K candidates with cosine similarity and then rerank them using our cosine + sparsity score function. We set a very large $K$ (e.g. $K = 1000$) so that empirically this is almost equivalent to searching for the maximal cosine + sparsity score in the whole corpus

To construct our training data, for each argument and counter-argument pair $(x_i, x_i^c)$ in the Arguana's training set, we set $x_i^c$ to be the positive example of $x_i$. We select all the other arguments and counter-arguments from the same debate and stance as $x_i$'s hard negatives. We fine-tune three pretrained sentence embedding models of different sizes ("UAE-Large-V1" (Li & Li, 2023), "GTE-large-en-v1.5" (Li et al., 2023), and "bge-base-en-v1.5" (Xiao et al., 2023)). Please refer to Table 10 for our training parameters.

**Baselines** We are not aware of any accurate methods for retrieving contradictions that only rely on sentence embeddings. To the best of our knowledge, we provide two baselines in our main experiment:

- CL: standard contrastive learning with cosine similarity, using the same training data (contradictions as positive examples and paraphrases as negative examples) that we use for our SPARSECL.

- Prompt + CL: standard contrastive learning with prompt "Not true: " attached in front of the query during both training and testing.

We report the performance of several efficient (with fewer than 1B parameters) and top-ranked pretrained sentence embedding models including "GTE-large-en-v1.5", "UAE-Large-V1", "bge-base-en-v1.5",

**Test** The Arguana test set consists of 1401 query arguments and counter-argument pairs. Following the standard test setting, we search for an answer of a query within the whole corpus (training set + validation set + test set) and report NDCG@10 scores. We select $\alpha$ based on the best NDCG@10 score on the validation set. When we directly use a model to provide cosine similarity scores in a zeroshot manner, we use its default pooler ("cls") for that model. When we use a fine-tuned model (via either CL or SPARSECL) to provide either cosine similarity scores or sparsity scores, we use the "avg" pooler.

**Results** The detailed results are presented in Table 1. Across all models—"GTE-large-en-v1.5", "UAE-Large-V1", and "bge-base-en-v1.5"—an average improvement of 3.6% in counter-argument retrieval were observed when incorporating our SPARSECL to either Zeroshot or CL. Furthermore, our CL (Cosine) + SPARSECL (Hoyer) method achieves NDCG@10 score 81.3 using GTE with only 400M parameters. For completeness, we also compare our results with (Shi et al., 2023) in Appendix C.

This pattern of enhancement was consistently observed regardless of whether the embedding models were fine-tuned or not. Notably, standard cosine similarity fine-tuning alone

also contributed to performance gains. For instance, fine-tuned GTE models showed an increase from 72.5 to 77.8 on the Arguana dataset using standard cosine similarity alone. This suggests that the Arguana dataset inherently favors scenarios where the counterargument is the most similar passage to the query, which may amplify the benefits of fine-tuning.

Interestingly, for the "Prompt + CL (Cosine)" method, fine-tuning with the appended prompt even results in a performance drop. During the training process, we observed overfitting and hypothesize that the special prompt "Not true:" introduces a shortcut, making it easier for the model to learn whether a text belongs to the "argument" class or the "counter-argument" class. However, this class information is not useful when identifying pairwise contradiction relationships.

These findings highlight the robustness of our approach, particularly when traditional similarity metrics are augmented with sparsity measures to capture subtle nuances in contradiction. Further insights can be gleaned from our ablation study detailed in Section 4.6, where we analyze the impact of similar non-contradictory passages within the corpus.

### 4.2. Contradiction retrieval

The task of "contradiction retrieval" generalizes beyond the argument and counter-argument relationship in the debate area, e.g. passages with conflicting factual information should also be considered as "contradictions". To test our method's validity for these more general forms of contradictions, we construct two datasets to test our method's performance.

**Data set construction** Given a QA retrieval dataset, e.g. MSMARCO (Nguyen et al., 2016), for each answer passage $x_i$ of a query $q_i$, we use Large Language Models (specifically, GPT-4 (Achiam et al., 2023)) to generate 3 answers paraphrasing $x_i$ or contradicting $x_i$. Let the generated paraphrases be $\{x_{i1}^+, x_{i2}^+, x_{i3}^+\}$ and the generated contradictions be $\{x_{i1}^-, x_{i2}^-, x_{i3}^-,\}$. We then delete $x_i$ from the corpus and add the set of generated passages $\{x_{i1}^+, x_{i2}^+, x_{i3}^+, x_{i1}^-, x_{i2}^-, x_{i3}^-\}$ to the corpus. In the test phrase, the queries are $\{x_{i1}^+, x_{i2}^+, x_{i3}^+\}$, each of which has the same answers $\{x_{i1}^-, x_{i2}^-, x_{i3}^-,\}$. We generate the paraphrases and contradictions for the validation set, test set, and a randomly sampled 10000 documents from the training set. Please refer to Appendix G for details.

**Training** To prepare the training data for contrastive learning, for each paraphrase and contradiction set $\{x_{i1}^+, x_{i2}^+, x_{i3}^+, x_{i1}^-, x_{i2}^-, x_{i3}^-\}$ generated from the same original passage, we form 9 pieces of training data $(x_{ia}^+, x_{ib}^-, x_{ic}^+)$ for 9 different combinations of paraphrases, contradictions,

| Model | Method | Arguana | MSMARCO | HotpotQA |
|-------|--------|---------|---------|----------|
| BGE | Zeroshot (Cosine) | 65.8 | 60.0 | 59.5 |
|  | CL (Cosine) | 68.7 | 52.7 | 56.2 |
|  | Prompt + CL (Cosine) | 64.4 | 61.1 | 80.4 |
|  | Zeroshot (Cosine) + **SPARSECL(Hoyer)** | 70.4 | **90.9** | **96.7** |
|  | CL (Cosine) + **SPARSECL(Hoyer)** | **72.2** | 88.3 | 96.5 |
| UAE | Zeroshot (Cosine) | 68.3 | 59.7 | 58.7 |
|  | CL (Cosine) | 70.4 | 44.2 | 54.1 |
|  | Prompt + CL (Cosine) | 63.7 | 85.8 | 94.3 |
|  | Zeroshot (Cosine) + **SPARSECL(Hoyer)** | 74.3 | **90.2** | **95.5** |
|  | CL (Cosine) + **SPARSECL(Hoyer)** | **74.4** | 86.9 | 94.3 |
| GTE | Zeroshot (Cosine) | 72.5 | 60.3 | 59.7 |
|  | CL (Cosine) | 77.8 | 65.1 | 59.7 |
|  | Prompt + CL (Cosine) | 71.1 | 88.1 | 69.0 |
|  | Zeroshot (Cosine) + **SPARSECL(Hoyer)** | 79.7 | **95.3** | 97.7 |
|  | CL (Cosine) + **SPARSECL(Hoyer)** | **81.3** | 95.2 | **97.9** |

*Table 1.* Results for different models and methods on the contradiction retrieval task. Experiments are run on the Arguana dataset (Wachsmuth et al., 2018) and modified MSMARCO(Nguyen et al., 2016) and HotpotQA(Yang et al., 2018) datasets. We report NDCG@10 score here, the higher the better. "UAE" stands for "UAE-Large-V1", "BGE" stands for "bge-base-en-v1.5", "GTE" stands for "gte-large-en-v1.5", The "Method" column denotes the score function used to retrieve contradictions. We consider two score functions: cosine similarity and cosine similarity plus Hoyer sparsity. "Zeroshot" denotes the direct testing of the model without any fine-tuning. "CL" denotes fine-tuning using standard contrastive learning. "Prompt+CL" denotes fine-tuning using standard contrastive learning with prompt "Not true:" attached in front of the query. "SPARSECL" denotes fine-tuning using Hoyer sparsity contrastive learning (our method).

and a randomly selected hard negative from the remaining two paraphrases. We then perform SPARSECL to fine-tune a sparsity-enhanced embedding.

**Test** Similar to the testing strategy for Arguana, we define our corpus to consist of all generated text (training set + validation set + test set). We query the paraphrases $\{x_{i1}^+, x_{i2}^+, x_{i3}^+\}$ of the original passage $x_i$ and set the groundtruth answers to be the generated contradictions $\{x_{i1}^-, x_{i2}^-, x_{i3}^-\}$. We select the $\alpha$ parameter with the maximal NDCG@10 score on the validation set and report the NDCG@10 score obtained by applying that $\alpha$ to the test set.

**Results** The results are reported in Table 1. For both MSMARCO and HotpotQA data sets, incorporating our SPARSECL method achieves over 14.6% percentage points gain compared with the two baselines. The large improvement is due to the existence of paraphrases in the corpus, that are strong confounders for the pure similarity-based methods.

We also observe that Prompt + CL (Cosine) performs much better on the MSMARCO and HotpotQA datasets compared to standard CL (Cosine). We propose two potential reasons: 1. We have 12 times more paraphrase and contradiction pairs generated for fine-tuning, which makes the model less likely to overfit. 2. The contradictions generated by GPT-

4 rely too heavily on opposite word replacement, which is better captured by the prompt 'not true.' However, the counter-arguments in the Arguana dataset use entirely different wording.

### 4.3. Zero-shot Generalization Test

To evaluate the generalization capability of our sparse-aware embeddings, we also conduct zero-shot tests on other datasets. Specifically, we train the embeddings on our synthetic HotpotQA or MSMARCO datasets and then test them on the other dataset in a zero-shot manner. We have confirmed that there is no data overlap between the two datasets. Please refer to Table 14 for the corresponding statistics. As presented in Table 2, SparseCL trained on MSMARCO or HotpotQA produces reasonable test results on the other dataset, albeit with a slight performance drop. Furthermore, using the same $\alpha$ parameter selected on the other dataset also gives reasonable test accuracy, showing the stability of $\alpha$ parameter across different datasets. This demonstrates that the sparse-aware embeddings trained on one dataset can capture contradiction relationships and generalize to unseen datasets.

### 4.4. Retrieval Corpus Cleaning

As an application of contradiction retrieval, we test how well our method can be used to find inconsistencies within a corpus and clean the corpus for future training or QA re-

| Method | Train Dataset | Test Dataset | NDCG@10 |
|---|---|---|---|
| Zeroshot(Cosine) | MSMARCO | HotpotQA | 88.1 |
| +SparseCL(Hoyer) +fixed $\alpha$ | HotpotQA | MSMARCO | 82.2 |
| Zeroshot(Cosine) | MSMARCO | HotpotQA | 88.6 |
| +SparseCL(Hoyer) + tuned $\alpha$ | HotpotQA | MSMARCO | 87.7 |
| Zeroshot(Cosine) | HotpotQA | HotpotQA | 96.7 |
| +SparseCL(Hoyer) | MSMARCO | MSMARCO | 90.9 |
| Zeroshot(Cosine) | N/A | HotpotQA | 59.5 |
| | N/A | MSMARCO | 60.0 |

*Table 2.* Results for zero-shot generalization experiment for contradiction retrieval running on "bge-base-en-v1.5" model

trieval. We first inject corrupted data contradicting existing documents into the corpus, and measure the retrieval accuracy degradation for retrieved answers. Then, we use our contradiction retrieval method to filter out corrupted data and measure the retrieval accuracy again.

**Data**   Similarly to the data generation in Section 4.2, we construct a new corpus containing LLM-generated paraphrases and contradictions based on MSMARCO and HotpotQA data sets. We start with an original corpus $C$ and its subset $S$. We then generate paraphrases and contradictions for $S$ as in Section 4.2.

For HotpotQA, $S$ contains all answer documents for the test set, 10000 answer documents sampled from the training set, and 1000 answer documents sampled from the development set. For MSMARCO, $S$ contains all answer documents for the dev set, and 11000 answer documents sampled from the training set.

We then curate 3 different versions of the corpus based on the original corpus $C$ and the subset $S$.

- The initial corpus $C^+$: For each original answer document $x$ in $S$, we remove $x$ from $C$ and instead add 3 LLM-generated paraphrases $\{x_1^+, x_2^+, x_3^+\}$ to $C$. The result forms the *initial* corpus $C^+$.

- The corrupted corpus $C^-$: For each original answer document $x$ in $S$, we generate 3 contradictions $\{x_1^-, x_2^-, x_3^-\}$ and add them to $C^+$ to get the *corrupted* corpus $C^-$.

- The cleaned corpus $C^\natural$: We apply our data cleaning procedure to the corrupted corpus $C^-$, obtaining the *cleaned* dataset $C^\natural$.

**Test**   We test the retrieval accuracy (NDCG@10) and the corruption ratio (Recall@10) for answering the original queries in the test set. The goal of our experiment is to show how retrieval algorithms behave on these three constructed corpora $C^+$, $C^-$, and $C^\natural$.

**Data Cleaning**   Our sparsity-based method can only identify contradictions within the data set, but we do not know which element in a contradiction pair is correct. To perform data cleaning, we make the assumption that for each original passage $x \in S$, we are given one of its paraphrases as the groundtruth. Then, our task is reduced to searching for passages contradicting a given ground truth document and filtering them out.

**Method**   We use the GTE-large-en-v1.5 model without fine-tuning to provide the cosine similarity score for this data cleaning experiment. We use the model from our contradiction retrieval experiment in section 4.2 trained on MS-MARCO and HotpotQA to provide the sparsity score. The $\alpha$ parameter is also identical to the one used in section 4.2. For each ground truth document, we filter out the top 3 scored documents from the corpus.

| Datasets | Original Acc | Corrupted Acc | Corrupt | Cleaned Acc | Corrupt |
|---|---|---|---|---|---|
| HotpotQA | 67.6 | 56.7 | 44.3 | 65.2 | 2.0 |
| MSMARCO | 43.5 | 38.1 | 41.3 | 41.4 | 4.0 |

*Table 3.* Experimental results for the impact of corrupted data on QA retrieval and contradiction retrieval for filtration. "Acc" represents the retrieval accuracy measured by the NDCG@10 score and "Corrupt" represents the fraction of returned passages that are corrupted, as measured by Recall@10.

Table 3 shows the results. We observe that the retrieval accuracy on the corrupted corpus drops significantly, as the generated contradictions cause the embedding model to retrieve them as query answers. The corruption ratio measures the average fraction of the top-10 retrieved documents that correspond to the generated contradicting passages. This performance is above 40% for both datasets. After performing our corpus cleaning procedure, which searches for the passages contradicting the given ground truth documents and removes the top-3 for each of them, we can recover more than 60% of the performance loss due to corruption and at the same time reduce the corruption ratio to less than 5%.

### 4.5. Score Functions for Natural Language Inference

As an application of our SPARSECL method, we demonstrate that our method can be useful for distinguishing contradictions from entailments and random pairs in natural language inference datasets. For SNLI (Bowman et al., 2015) and MNLI (Williams et al., 2018) datasets, we extract entailment and contradiction pairs, fine-tune using standard contrastive learning and our SPARSECL, and then report the average cosine similarity / Hoyer sparsity score between entailments, contradictions, and random pairs.

We can observe from Table 5 that, in the zeroshot setting, the

| Model | Method | $l_2/l_1$ | $\kappa_4$ | Hoyer | Cosine (baseline) |
|-------|--------|-----------|-----------|-------|-------------------|
| BGE | Zeroshot (Cosine) + SPARSECL | 67.5 | 68.4 | **70.4** | 65.7 |
| BGE | CL (Cosine) + SPARSECL | 70.2 | 70.7 | **72.2** | 68.7 |

*Table 4.* NDCG@10 scores for Arguana using SPARSECL with different sparsity functions. We also report two baselines that use only the cosine similarity (zeroshot and contrastive learning).

| | | Contradiction | Entailment | Random |
|------|-----|---------------|------------|--------|
| | Zeroshot (Cosine) | 54.6 | 76.9 | 37.6 |
| SNLI | CL (Cosine) | 88.5 | 88.6 | 77.7 |
| | SparseCL (Hoyer) | **37.6** | **34.7** | **22.8** |
| | Zeroshot (Cosine) | 65.9 | 81.8 | 37.8 |
| MNLI | CL (Cosine) | 91.9 | 91.7 | 73.3 |
| | SparseCL (Hoyer) | **42.2** | **36.4** | **24.4** |

*Table 5.* Average Cosine / Hoyer scores between Contradiction / Entailment / Random pairs of texts. The experiment is run on "bge-base-en-v1.5" model. Texts pairs are from SNLI and MNLI datasets

average cosine similarity of contradiction pairs lies between the ranges of random and entailment pairs. For the fine-tuned model using standard contrastive learning (CL), the average cosine similarity of contradiction pairs is almost indistinguishable from that of entailment pairs. Finally, after being fine-tuned using SPARSECL, the model exhibits higher average Hoyer sparsity scores for contradiction pairs compared to other two types of relationships.

### 4.6. Ablation Studies

We perform the following three ablation studies to further understand sparsity-based retrieval method.

**Arguana retrieval results analysis**    In the standard Arguana dataset, even though the task is to retrieve the counter-argument for the query, the retrieval based solely on similarity still gives reasonable results. This means that counter-arguments are also the most similar arguments to the query, which makes the data set an imperfect test bed for testing contradiction retrieval.

To further compare our sparsity-based method and the pure similarity-based method , we augment Arguana by adding arguments' paraphrases to the corpus. Specifically, for any argument $x$ and its counter-argument $x^-$ in the original corpus $C$, we use GPT-4 to generate three paraphrases $\{x_1, x_2, x_3\}$ of $x$. We then form three new corpora with an increasing number of paraphrases added to the corpus: $C_1$ contains all $x_1$ and $x^-$, $C_2$ contains all $x_1, x_2$, and $x^-$, and $C_3$ contains all $x_1, x_2, x_3$, and $x^-$.

In the testing phase, we query the counter-arguments for one of $x$'s paraphrases, the answer of which should still be $x^-$. We observe how the performance varies when the corpora

we retrieve from are $C_1, C_2, C_3$.

| Methods | $C_1$ | $C_2$ | $C_3$ |
|---------|-------|-------|-------|
| Zeroshot (Cosine) | 56.1 | 35.5 | 26.7 |
| **Zeroshot (Cosine) + SPARSECL(Hoyer)** | 68.2 | 67.9 | 67.5 |
| CL (Cosine) | 47.1 | 30.3 | 22.8 |
| **CL (Cosine) + SPARSECL(Hoyer)** | 61.9 | 61.8 | 61.5 |

*Table 6.* Counter-argument retrieval results on the augmented Arguana dataset with different numbers of similar arguments in the corpus. $C_x$ denotes testing counter-argument retrieval on the corpus with $x$ existing paraphrases (including itself) of the query argument. Experiments were run on "bge-base-en-v1.5" model.

We present our overall experimental results in Table 6. Please also refer to Appendix D for an example case study. As the number of paraphrases in corpus increases from 1 to 3, the performance of the similarity-based method drops significantly. Thus it is reasonable to deduce that, as the number of similar arguments in the corpus increases further, the NDCG@10 scores for similarity-based methods will converge to 0. On the other hand, the performance of our sparsity-based method is stable with respect to the number of paraphrases in the corpus.

**Different sparsity functions**    Our intuition in Section 3 does not give clear guidelines on which sparsity function to use in our SPARSECL. Thus, we also experiment with different choices of sparsity functions, selected from Hurley & Rickard (2009). Specifically, we consider two other sparsity functions ($l_2/l_1$ and $\kappa_4$), which are scale invariant and differentiable (see Table III in (Hurley & Rickard, 2009)). Note that both of these two sparsity functions have ranges $[0, 1]$, and higher values of those functions correspond to sparser vectors.

$$\frac{l_2}{l_1} = \frac{\|h_1 - h_2\|_2}{\|h_1 - h_2\|_1} \qquad \kappa_4 = \frac{\|h_1 - h_2\|_4^4}{\|h_1 - h_2\|_2^2}.$$

As per Table 4, compared to the cosine similarity method, the combination of the cosine similarity score with the sparsity score trained by SPARSECL, yields higher NDCG@10 scores for each sparsity function. However, Hoyer sparsity yields the highest accuracy. We believe that simple sparsity functions have a more benign optimization landscape and thus are easier for models to optimize.

**Different retrieval methods for contradiction retrieval**
We experiment with 5 retrieval methods in our ablation study. The methods evaluated are as follows: "Prompt" involves appending the "Not true: " prompt to the query during testing, followed by standard similarity search. "Prompt + CL (Cosine)" extends this by incorporating contrastive learning with the "Not true: " prompt included in the training data. "Gen" uses GPT-4 to generate contradictions to the query (details in Appendix G) and applies similarity search for testing. "Gen + CL (Cosine)" fine-tunes using contrastive learning with the generated contradictions in the training data before similarity search. Finally, "SparseCL (Hoyer)" employs SparseCL fine-tuning and retrieves documents based on the maximal Hoyer sparsity score during testing.

As shown in Table 7, we observe that generally "Gen" and "Prompt" don't improve much upon standard similarity search. For the "Gen + CL (Cosine)" method, a diverse set of counter-arguments exist for a given argument, making it hard to generate a single counter-argument that closely matches the true ground truth counter-argument. For the "Prompt + CL (Cosine)" method, fine-tuning with the appended prompt even results in a performance drop. During the training process, we observed overfitting and hypothesize that the special prompt "Not true:" introduces a shortcut, making it easier for the model to learn whether a text belongs to the "argument" class or the "counter-argument" class. However, this class information is not useful when identifying pairwise contradiction relationships. Finally, directly using Hoyer sparsity to retrieve contradictions doesn't yield good results as well, because we believe contradictions involve a combination of similarity and dissimilarity.

| Model | Method | Arguana |
|-------|--------|---------|
| BGE | Prompt + Zeroshot (Cosine) | 65.7 |
|  | Gen + Zeroshot (Cosine) | 64.7 |
|  | Zeroshot (Cosine) | 65.8 |
| BGE | Prompt + CL (Cosine) | 64.5 |
|  | Gen + CL (Cosine) | 70.0 |
|  | CL (Cosine) | 68.7 |
| BGE | SparseCL (Hoyer) | 56.1 |
|  | CL (Cosine) + SparseCL (Hoyer) | 72.2 |

*Table 7.* Counter-argument retrieval results (NDCG@10 scores) on Arguana dataset with different retrieval methods. "Gen" means using GPT-4 to generate a contradiction $c$ of the query argument $q$, "Prompt" means appending the "Not true : " prompt in the front of the query text. "Zeroshot" refers to direct testing and "CL" and "SparseCL" refer to fine-tuning with respective methods.

## 5. Conclusion

In this work, we introduced a novel approach to contradiction retrieval that leverages the sparsity between sentence embeddings, combined with cosine similarity, to efficiently identify contradictions in large document corpora. This method addresses the limitations of the traditional similarity search as well as computational inefficiencies of the cross-encoder models, proving its effectiveness on benchmark datasets like Arguana and on synthetic contradictions retrieval from MSMARCO and HotpotQA.

## Acknowledgements

Haike Xu was supported by the Mathworks Fellowship. Piotr Indyk was supported in part by the NSF TRIPODS program (award DMS-2022448). Zongyu Lin was partially supported by DARPA ANSR program FA8750-23-2-0004, NSF 2331966 NSF 2211557, NSF 2119643, NSF 2303037, NSF 2312501, SRC JUMP 2.0 Center, Amazon Research Awards, and Snapchat Gifts.

## Impact Statement

This paper introduces a novel approach to the contradiction retrieval task that relies solely on text embeddings. To the best of our knowledge, we are the first to consider such non-similarity-based retrieval problem. We consider this an important yet underexplored area and hope our work will spark greater interest in this field.

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

# A. Additional related work

**Complex retrieval tasks**   Information retrieval is a well-studied area (Singhal et al., 2001) and there have been many benchmarks for testing retrieval performance such as BEIR (Thakur et al., 2021), MTEB (Muennighoff et al., 2023), and MIRACL (Zhang et al., 2023a). However, most of the datasets, through varying in some degrees, focus only on "retrieving the most similar document". People have noted that there exist some more complex retrieval tasks (e.g. Arguana (Wachsmuth et al., 2018) retrieves counter-arguments that refute a query argument), and build retrieval benchmark focusing on complex retrival goals, e.g. BIRCO (Wang et al., 2024) and BERRI (Asai et al., 2023).

To retrieve according to different instructions, (Asai et al., 2023) trains TART, a multi-task retrieval system with task instructions attached as prompts in front of the query content. However, when answering queries, they are still searching for the most similar sentence embedding, though the prompt is different for different tasks. As far as we know, our paper studies the first non-similarity-based search problem.

**Data inconsistency and misinformation detection**   Data inconsistency, refers to the factually incorrectness in the content, might come from different sources, including their natural existence in the corpus (Shahi & Nandini, 2020; Cui & Lee, 2020), data augmentations (Jha et al., 2020; Zhou et al., 2022), and pseudo labeling (Xie et al., 2020; Wang et al., 2022), which might lead to negative influence if serving as training dataset. There have been a few datasets on detecting the factually wrong information. For example, (Laban et al., 2022) detects whether a given summary is consistent with the input document, (Shahi & Nandini, 2020; Cui & Lee, 2020) detects whether a given COVID-19 related news is true or false. Most of these datasets lie in a specific domain and require external knowledge to judge the correctness of each piece of data. On the contrary, the "data inconsistency" notion we consider in our paper doesn't depend on any external knowledge, but is a relationship between different pieces of data in the same corpus. The goal of our method is to find such "contradiction pairs" in corpus efficiently, but not to judge which one is consistent with the real world knowledge.

# B. The "non-transitivity" of Hoyer sparsity and the "transitivity" of cosine function

Here, we provide a simple example to demonstrate that using Hoyer sparsity to measure "contradiction" can bypass the challenging scenario for similarity metrics where "A contradicts C, B contradicts C, but A doesn't contradict B". Specifically, Hoyer sparsity satisfies the following "non-transitivity" property.

**Proposition B.1** ("non-transitivity" of hoyer sparsity). *There exist three vectors A, B, and C of dimensionality d, satisfying* $1 \leq \|A\|_2, \|B\|_2, \|C\|_2 \leq 1 + O(\frac{1}{\sqrt{d}})$, *such that* $Hoyer(A, C) > 1 - O\left(\frac{1}{\sqrt{d}}\right)$, $Hoyer(B, C) > 1 - O\left(\frac{1}{\sqrt{d}}\right)$, *and* $Hoyer(A, B) < O(\frac{1}{\sqrt{d}})$

*Proof.* We construct the following $d$ dimensional vectors where $\epsilon < \frac{1}{d}$ can be any parameter.

$$
\begin{array}{rcllllll}
A & = & (1, & 0, & 0, & \ldots, & 0) \\
B & = & (1, & 0, & \epsilon, & \ldots, & \epsilon) \\
C & = & (0, & 1, & 0, & \ldots, & 0)
\end{array}
$$

Then, we calculate their $l_1$ over $l_2$ ratios:

$$
\frac{\|A - B\|_1}{\|A - B\|_2} = \sqrt{d - 2}
$$

$$
\frac{\|A - C\|_1}{\|A - C\|_2} = \sqrt{2}
$$

$$
\frac{\|B - C\|_1}{\|B - C\|_2} = \frac{2 + (d - 2)\epsilon}{\sqrt{2 + (d - 2)\epsilon^2}} < \frac{3}{\sqrt{2}}
$$

Applying their $l_1$ over $l_2$ ratio bounds to the Hoyer sparsity formula will give us the desired relationship.

$\square$

Next, we provide another example to demonstrate that the cosine function exhibits the following "transitivity" property, which makes it hard to characterize the scenario where "A contradicts C, B contradicts C, but A doesn't contradict B".

**Proposition B.2** ("transitivity" property of cosine function). *Given three unit vectors A, B, and C, if $cos(A, C) \geq 1 - O(\epsilon)$ and $cos(B, C) \geq 1 - O(\epsilon)$, we have $cos(A, B) \geq 1 - O(\epsilon)$*

*Proof.* For any two vectors $X$ and $Y$ with unit norm, we have $cos(X, Y) = 1 - \frac{\|X-Y\|_2^2}{2}$. Because $cos(A, C) \geq 1 - O(\epsilon)$, we have $\|A - C\|_2 \leq O(\sqrt{\epsilon})$. Finally, $cos(A, B) = 1 - \frac{\|A-B\|_2^2}{2} \geq 1 - \frac{(\|A-C\|_2 + \|C-B\|_2)^2}{2} \geq 1 - O(\epsilon)$ $\square$

## C. Experiment comparison with method from (Shi et al., 2023)

Shi et al. (2023) proposes "Bipolar-encoder" method to retrieve contradictions from the corpus. They also tested their method on the Arguana dataset but used a different metric, Recall@1. For completeness, we have translated our results into their Recall@1 metric for a fair comparison. As shown in Table 8, both our CL (baseline method) and CL+SparseCL (our method) demonstrate significant improvement over the previous results in (Shi et al., 2023).

| Model | Method | Arguana(Recall@1) |
|---|---|---|
| GTE | CL+SparseCL (ours) | 62.9 |
| GTE | CL (baseline) | 56.3 |
| (Shi et al., 2023) | Bipolar-encoder | 49.0 |

*Table 8.* Comparison of experimental results on the Arguana dataset

## D. A case study for counter-argument retrieval from Arguana dataset

In this section we provide an example to illustrate how our sparsity-based retrieval method is better at retrieving counter-arguments. In the setting of the augmented Arguana dataset (see our ablation study in Section 4.6), we selected an example query with an ID "aeghh-pro03a", for which we list the top 10 retrieved passages using the standard cosine similarity score and our sparsity-based score ($\alpha = 1.78$ selected from the dev set). The first five letters of a passage ID represent the argument topic ID; "pro/con" denotes the argument stance; suffix "a/b" indicates the argument and its corresponding counter-argument; "para0/para1/para2" are three paraphrases generated by GPT4.

As shown in Table 9, for the example query "aeghh-pro03a", its correct counter-argument, "aeghh-pro03b" (in red), ranks fourth using the cosine score but first using the cosine + hoyer score. Meanwhile, its paraphrases "aeghh-pro03a-para0/1/2" (in blue) achieve high cosine scores but low sparsity scores.

| Method | CL(Cosine) | | CL(Cosine)+SparseCL(Hoyer) | | | |
|---|---|---|---|---|---|---|
| Rank | Cosine | Passage ID | Overall | Cosine | Hoyer | Passage ID |
| 1 | 0.940 | **aeghh-pro03a-para0** | 1.683 | 0.794 | 0.499 | **aeghh-pro03b** |
| 2 | 0.926 | **aeghh-pro03a-para2** | 1.644 | 0.719 | 0.519 | aeghh-con02a-para0 |
| 3 | 0.916 | **aeghh-pro03a-para1** | 1.617 | 0.716 | 0.506 | aeghh-con02a-para2 |
| 4 | 0.794 | **aeghh-pro03b** | 1.606 | 0.940 | 0.374 | **aeghh-pro03a-para0** |
| 5 | 0.719 | aeghh-con02a-para0 | 1.602 | 0.718 | 0.496 | aeghh-con02a-para1 |
| 6 | 0.718 | aeghh-con02b | 1.528 | 0.718 | 0.454 | aeghh-con02b |
| 7 | 0.718 | aeghh-con02a-para1 | 1.494 | 0.916 | 0.324 | **aeghh-pro03a-para1** |
| 8 | 0.716 | aeghh-con02a-para2 | 1.426 | 0.926 | 0.280 | **aeghh-pro03a-para2** |
| 9 | 0.696 | aeghh-con02a | 1.396 | 0.669 | 0.408 | dhwif-pro02b |
| 10 | 0.692 | aeghh-pro04a-para0 | 1.344 | 0.628 | 0.402 | thggl-con03b |

*Table 9.* An example query analysis for counter-argument retrieval. The passage ID in red represents the ground-truth counter-argument, while the passage IDs in blue are paraphrases of the query argument.

# E. Hyper-parameters for training and inference

Here we present the training details (Table 10) for our experiments on Arguan, HotpotQA, and MSMARCO.

| Models | Model Size | Backbone | CL | | SPARSECL | | temp | bz |
|---|---|---|---|---|---|---|---|---|
| | | | ep | lr | ep | lr | | |
| GTE-large-en-v1.5 | 434M | BERT + RoPE + GLU | 1 | 1e-5 | 3 | 2e-5 | 0.01 | 64 |
| UAE-Large-V1 | 335M | BERT | 1 | 2e-5 | 3 | 2e-5 | 0.02 | 64 |
| bge-base-en-v1.5 | 109M | BERT | 1 | 2e-5 | 3 | 2e-5 | 0.02 | 64 |

*Table 10.* Training parameters for Arguana. We set max sequence length to be 512 for Arguana dataset and 256 for HotpotQA and MSMARCO datasets.

# F. Efficiency test of cross-encoder and vector calculation

To further compare the efficiency of cross-encoders and Hoyer sparsity calculations, we perform the following experiments:

- We choose "bge-reranker-base" and "bge-reranker-large" to be our cross-encoders. We use them to calculate the similarity between one query from Arguana's test set and 100 documents from Arguana's corpus. We report the average running time of this method for 100 queries.

- We choose "bge-base-en-v1.5" and "bge-large-en-v1.5" to be our bi-encoders. Suppose we have preprocessed all the sentence embeddings. We use it to calculate the Hoyer sparsity between one query embedding from Arguana's test set and 100 document embeddings from Arguana's corpus. We report the average running time of this method for 100 queries.

Please see Table 11 for the running time of different methods. We can see that the calculation of Hoyer sparsity is at least 200 times faster than running a cross-encoder.

| Cross-encoder | Model size | Time |
|---|---|---|
| bge-reranker-base | 278M | 0.8832s |
| bge-reranker-large | 560M | 1.6022s |
| Bi-encoder | Embedding dimension | Time |
| bge-base-en-v1.5 | 768 | 0.0029s |
| bge-large-en-v1.5 | 1024 | 0.0036s |

*Table 11.* Average running time for calculating the score functions between one Arguana query and 100 Arguana documents

# G. Data generation details for MSMARCO and HotpotQA experiments in Section 4.2

We use "gpt-4-turbo" to generate paraphrases and contradictions for our experiment in Section 4.2. The prompts we use are in Table 13. We set $temperature = 1$ and $n = 3$ (to generate 3 outputs). Please see Table 12 for some examples of generated paraphrases and contradictions and Table 14 for the number of unique passages generated from each dataset split. We have verified that there is no overlap between different splits or different dataset. For generated data quality, we sampled 500 data points from the dataset and 2 people annotated the data to check the quality. Our agreement score (using Cohen's Kappa) is 0.98, indicating the quality of the generated data. We present the first 20 passages and their generated paraphrases and contradictions from MSMARCO and HotpotQA datasets in the end of this section for readers' reference.

| Datasets | Orginal | Paraphrase | Contradiction |
|---|---|---|---|
| MSMARCO | In addition to the **high financial value** of higher education, higher education also makes individuals much **more intelligent** than what they would be with just a high school education... | Beyond its **significant monetary worth**, higher education substantially **enhances a person's intelligence** compared to merely completing high school... | Besides the **low financial significance** of higher education, higher education often renders individuals **no more intelligent** than they would be with just a high school education... |
| HotpotQA | Ice hockey is a **contact** team sport played on **ice**, usually in a **rink**, in which two teams of skaters use their **sticks** to shoot a **vulcanized rubber puck** into their opponent's net to score points... | Ice hockey is a **contact** sport where two teams compete on an **ice surface**, typically in a **rink**, using **sticks** to hit a **vulcanized rubber puck** into the opposing team's net to earn points... | Ice hockey is a **non-contact** team sport played on **grass**, often in an **open field**, where two teams of players use their **feet** to kick a **soft leather ball** into their opponent's goal to score points... |

*Table 12.* Examples of passages from MSMARCO and HotpotQA datasets, with their generated paraphrases, and generated contradictions. Highlighted key-words represent exact matchings or contradictions

| Task | Prompt |
|---|---|
| Generating paraphrases | Paraphrase the given paragraph keeping its original meaning. Do not add information that is not present in the original paragraph. Your response should be as indistinguishable to the original paragraph as possible in terms of length, language style, and format. Begin your answer directly without any introductory words. |
| Generating contradictions | Rewrite the given paragraph to contradict the original content. Ensure the revised paragraph changes the factuality of the original. Your response should be as indistinguishable to the original paragraph as possible in terms of length, language style, and format. Begin your answer directly without any introductory words. |

*Table 13.* Prompts used to generate paraphrases and contradictions for MSMARCO and HotpotQA documents.

| Dataset | Train | Dev | Test |
|---|---|---|---|
| MSMARCO | 59500 | 5950 | 44101 |
| HotpotQA | 59074 | 5896 | 81673 |

*Table 14.* Number of unique passages generated from MSMARCO and HotpotQA corpus via GPT-4. We have verified that there is no overlap between different splits or different datasets.

| Passage ID | Orginal | Paraphrase | Contradiction |
|---|---|---|---|
| MSMARCO-6725993 | What is a BIOS? BIOS is an acronym for Basic Input / Output System. On virtually every computer available, the BIOS makes sure all the other chips, hard drives, ports and CPU function together. What BIOS Does The BIOS software has a number of different roles, but its most important role is to load the operating system. When you turn on your computer and the microprocessor tries to execute its first instruction, it has to get that instruction from somewhere. | BIOS stands for Basic Input/Output System. It exists on almost all computers and is responsible for ensuring that all the various components such as chips, hard drives, ports, and the CPU work in unison. The primary function of BIOS software is to initiate the operating system. Upon powering up your computer, the microprocessor seeks to perform its initial instruction, which it retrieves through the BIOS. | What is a BIOS? BIOS stands for Basic Input / Output System. On nearly no modern computer, the BIOS is used to allow all the other chips, hard drives, ports, and CPU to operate disjointedly. What BIOS Does Not Do The BIOS software has limited functions, and its least significant role is to load the operating system. When you turn on your computer and the microprocessor attempts to execute its first instruction, it does not rely on the BIOS to obtain that instruction. |
| MSMARCO-6909689 | Causes of nitrates in urine. 1 Nitrites/nitrates in urine refer to a byproduct formed due to the action of bacteria occurring in the urinary tract. The kidneys perform the vital function of cleaning the blood by filtering out the unnecessary, toxic matter from blood. The nitrite developed by bacteria cannot be filtered by the kidneys. | Reasons for nitrates in urine. 1 Nitrates or nitrites in urine are byproducts produced from bacterial activity within the urinary tract. The kidneys have the critical role of purifying the blood by removing unwanted, harmful substances. However, the nitrites created by such bacteria are not able to be filtered out by the kidneys. | Causes of nitrates in urine. 1 Nitrites/nitrates in urine signify a byproduct that is not associated with bacterial action in the urinary tract. The kidneys, unable to filter these components once formed, do not influence their presence in the urine. Instead, nitrites converted from nitrates in food enter the urinary system independent of any bacterial intervention. |
| MSMARCO-594175 | Food Additives. Food additives are substances added intentionally to foodstuffs to perform certain technological functions, for example to colour, to sweeten or to help preserve foods. In the European Union all food additives are identified by an E number. Food additives are always included in the ingredient lists of foods in which they are used. | Food Additives. Substances known as food additives are deliberately incorporated into foods to fulfill specific technological roles such as coloring, sweetening, or preserving the food. In the European Union, these additives are marked with an E number. The presence of food additives in products is consistently disclosed in the ingredient lists of the foods where they are utilized. | Food Additives. Food additives are substances naturally present in food items to perform specific biological purposes, such as to enhance flavor, to sour, or to decrease the shelf life of foods. In the European Union, no specific identification like an E number is utilized for food additives. Food additives are rarely disclosed in the ingredient lists of foods in which they appear. |
| MSMARCO-6208116 | monosubstituted alkene. (organic chemistry). An alkene with the general formula RHC=CH 2, where R is any organic group; only one carbon atom is bonded directly to one of the carbons of the carbon-to-carbon double bond.1 Facebook.2 Twitter. | monosubstituted alkene. (organic chemistry). An alkene characterized by the general formula RHC=CH2, where R represents any organic group; a single carbon atom is directly bonded to one carbon of the double bond between carbons.1 Facebook.2 Twitter. | disubstituted alkene. (organic chemistry). An alkene with the general formula RHC=CHR, where R represents any organic group; each carbon atom in the carbon-to-carbon double bond is directly bonded to its own distinct organic group. 1 Facebook. 2 Twitter. |

| Passage ID | Orginal | Paraphrase | Contradiction |
|---|---|---|---|
| MSMARCO-3366870 | Depending on the specific germ, the incubation period is between 12 hours and 5 days, usually 48 hours. To answer your question, the common cold is contagious between 24 hours before onset of symptoms until 5 days after on-set. | The incubation period varies from 12 hours to 5 days, typically around 48 hours, depending on the germ involved. Regarding your question, one can spread the common cold from 24 hours be-fore symptoms start up to 5 days following their appearance. | Depending on the specific germ, the incubation period is between 6 days and 2 weeks, usually 7 days. To answer your question, the common cold is contagious from 5 days before the onset of symptoms until 10 days after on-set. |
| MSMARCO-6494839 | Noun. 1. decrescendo - (music) a gradual decrease in loudness. diminuendo. softness-a sound property that is free from loud-ness or stridency; and in softness almost beyond hearing. music-an artistic form of auditory commu-nication incorporating instrumen-tal or vocal tones in a structured and continuous manner. | Noun. 1. decrescendo - (music) a progressive reduction in volume. diminuendo. softness - a charac-teristic of sound marked by low volume and an absence of harsh-ness; approaching a level that is barely audible. music - an art form of auditory communication that uses instrumental or vocal sounds in a structured and sus-tained way. | Noun. 1. crescendo - (mu-sic) a gradual increase in loud-ness. crescendo. loudness-a sound property characterized by high volume and stridency; and in loudness almost beyond en-durance. silence-a form of non-auditory communication devoid of instrumental or vocal tones, un-structured and intermittent. |
| MSMARCO-934134 | Due to its important role in curdling milk, rennin enzyme is widely used in the food in-dustry, notably in the produc-tion of cheese. Rennin for cheese-making was once derived mainly from the dried stomachs of calves and from some non-animal sources.ennin enzymes are produced by the stomach cells of young mammals. Rennin is secreted in large amounts right after the birth and then its pro-duction gradually drops off. It is then eclipsed in importance by the Pepsin enzyme. | Rennin enzyme, critical for milk curdling, is extensively utilized in the food sector, especially for cheese production. Histori-cally, cheese-making rennin was sourced primarily from the dried stomachs of young calves and var-ious non-animal origins. This en-zyme is secreted by the stomach cells of young mammals, with secretion levels peaking immedi-ately following birth before grad-ually declining. Over time, the significance of rennin decreases as it is overshadowed by the en-zyme Pepsin. | Despite its crucial function in cur-dling milk, the rennin enzyme is minimally used in the food industry, particularly in the pro-duction of cheese. Rennin for cheese-making used to be primar-ily sourced synthetically and not from natural origins such as the dried stomachs of calves. Ren-nin enzymes are produced by the stomach cells of older mammals. Rennin is secreted in minimal amounts right after birth, and then its production progressively in-creases. It continues to retain its importance over the Pepsin en-zyme. |
| MSMARCO-4473191 | Winter Moth (Operophtera bru-mata). up in traps, at least, in southeastern NH, coastal Maine, one place in southeastern CT and out on Long Island. Mas-sachusetts still appears to have the largest and most damaging pop-ulations of this pest. spanworm (Eidt et al. | Winter Moth (Operophtera bru-mata) has been caught in traps primarily in southeastern New Hampshire, coastal Maine, a loca-tion in southeastern Connecticut, and on Long Island. The most sig-nificant and harmful populations of this pest seem to still be in Mas-sachusetts, according to Eidt et al. | Summer Moth (Operophtera sol-sticia). down in nets, at least, in northwestern NH, inland Maine, one place in northwestern CT and out on Long Island. Mas-sachusetts seems to have the smallest and least harmful pop-ulations of this pest. inchworm (Eidt et al. |

| Passage ID | Orginal | Paraphrase | Contradiction |
|---|---|---|---|
| MSMARCO-7066556 | Actor Charles Bronson dies at 81 Death Wish movie star Charles Bronson, the coal miner from Pennsylvania who drifted into films as a villain and became a hard-faced action star, has died. Bronson, 81, died of pneumonia at Cedars-Sinai Medical Centre in Los Angeles, with his wife at his bedside, publicist Lori Jonas said. He had been in the hospital for weeks. | Charles Bronson, star of 'Death Wish' and former Pennsylvania coal miner who turned to acting as a villain before becoming a renowned action hero, has passed away at the age of 81. Bronson died from pneumonia at Cedars-Sinai Medical Centre in Los Angeles while his wife was by his side, his publicist Lori Jonas confirmed. He had been hospitalized for several weeks. | Actor Charles Bronson thrives at 81 Death Wish movie star Charles Bronson, the coal miner from Pennsylvania who drifted into films as a hero and became a beloved romantic lead, is thriving. Bronson, 81, recently recovered from a mild cold at his home in Los Angeles, without needing hospital care, publicist Lori Jonas said. He has been in excellent health. |
| MSMARCO-2539424 | (Redirected from Primary Wave Music) Primary Wave Entertainment is a full-service entertainment company with expertise in Talent Management, Literary Management, Music Publishing, Branding, Digital Marketing and Licensing based in the United States. | Primary Wave Entertainment is a comprehensive entertainment firm based in the United States, specializing in Talent Management, Literary Management, Music Publishing, Branding, Digital Marketing, and Licensing. | Secondary Echo Entertainment is a specialized entertainment company lacking capabilities in Talent Management, Literary Management, Music Publishing, Branding, Digital Marketing, and Licensing, located outside the United States. |
| MSMARCO-5893222 | As of 2010, median salaries for unit secretaries ranged from $ 23,098-$ 31,604, while median hourly wages for hospital unit secretaries ranged from $10.83-14.67$, according to PayScale.com. | In 2010, the median annual salaries for unit secretaries varied between $ 23,098 and $ 31,604, and their median hourly pay in hospitals was between $ 10.83 and $ 14.67, as reported by PayScale.com. | As of 2010, median salaries for unit secretaries varied from $ 35,000-$ 42,500, while median hourly wages for hospital unit secretaries were between $ 16.50-$ 19.75, according to PayScale.com. |
| MSMARCO-3476298 | Answers.com® is making the world better one answer at a time. cleffa evolves at level 30 you can find it at mt. coronet. 2 people found this useful. | Answers.com® improves the world by providing one answer at a time. Cleffa evolves at level 30 and can be found at Mt. Coronet. This information was useful to 2 people. | Answers.com® often misleads more than it informs. Cleffa evolves with high friendship, not at level 30, and it is found in multiple locations, not just Mt. Coronet. 2 people found this misleading. |

| Passage ID | Orginal | Paraphrase | Contradiction |
|---|---|---|---|
| MSMARCO-6150895 | In the sidebar, make sure you select Music under the iPod (indented below the iPod). That's the iPod's content list and shows the list of songs on the iPod. Do NOT select Music under LIBRARY. If you want to delete ALL of the songs on the iPod, click on any song on that song list.Do a Select All, which is Cmd-A on the keyboard or Select All from the Edit menu (in menu bar). With all of the songs selected, press Delete on the keyboard. | In the sidebar, choose Music located beneath the iPod (indented under the iPod). This displays the list of songs stored on your iPod. Be sure not to select Music listed under LIBRARY. To remove all songs from the iPod, click on any track in that music list. Perform a Select All by pressing Cmd-A on your keyboard or by choosing Select All from the Edit menu in the menu bar. Once all songs are highlighted, hit the Delete key on your keyboard. | In the sidebar, ensure that you choose Music under LIBRARY, not under the iPod (indented below the iPod). That selection links to your computer's music collection instead of displaying the list of songs stored on the iPod. Always select Music under LIBRARY. If you wish to preserve all the songs on the iPod, avoid clicking on any song in that song list. Do not use Select All, which would be Cmd-A on the keyboard or Select All from the Edit menu (in menu bar). With none of the songs selected, do not press Delete on the keyboard. Ensure you do not alter the content list that appears under the iPod selection in the sidebar. |
| MSMARCO-1814356 | The Associate of Applied Science Degree requires that you have completed the TSI requirements. Course Requirements Graduation with the Associate of Applied Science Degree in Welding or the completion of the Com-bination or Structural Welding Certificate re-quires successful completion of a Comprehensive Exit Exam. | The Associate of Applied Science Degree mandates fulfillment of TSI prerequisites. To graduate with an Associate of Applied Science Degree in Welding or to receive either the Combination or Structural Welding Certificate, it is essential to pass a Comprehensive Exit Exam. | The Associate of Applied Science Degree does not require you to have completed the TSI requirements. Course Requirements Graduation with the Associate of Applied Science Degree in Welding or the completion of the Combination or Structural Welding Certificate does not require the successful completion of a Comprehensive Exit Exam. |
| MSMARCO-6108145 | Pluto's rotation period, its day, is equal to 6.39 Earth days. Like Uranus, Pluto rotates on its side on its orbital plane, with an axial tilt of 120Â°, and so its seasonal variation is extreme; at its solstices, one-fourth of its surface is in continuous daylight, whereas another fourth is in continuous darkness.luto (minor-planet designation: 134340 Pluto) is a dwarf planet in the Kuiper belt, a ring of bodies beyond Neptune. | Pluto's day, its rotation period, spans 6.39 Earth days. Similar to Uranus, Pluto has a rotation along its orbital plane on its side, with an axial tilt of 120 degrees. Consequently, it experiences extreme seasonal changes; during its solstices, a quarter of its surface enjoys perpetual daylight while another quarter remains in constant darkness. Pluto, designated as minor-planet 134340, is classified as a dwarf planet located in the Kuiper belt, a collection of objects that orbits around Neptune. | Pluto's rotation period, its day, is equal to 153.3 Earth hours. Unlike Uranus, Pluto rotates upright in its orbital plane, with an axial tilt of 30°, and so its seasonal variation is moderate; at its solstices, nearly all of its surface experiences alternating daylight and darkness. Pluto (minor-planet designation: 134340 Pluto) is not considered a dwarf planet but a major planet, centrally situated in the Kuiper belt, a sparse ring of bodies beyond Neptune. |

| Passage ID | Orginal | Paraphrase | Contradiction |
|---|---|---|---|
| MSMARCO-5618382 | Domestic Costa Rica flights. Your premier Costa Rica airline choice for travel and vacation flights within Costa Rica, offering 74 daily flights to 17 destinations in Costa Rica! Nature Air the Costa Rica Domestic Airline, offers domestic flights to 17 destinations in Costa Rica. Nature Air Costa Rica Airline, offers 74 daily domestic flights including to and from Juan Santamaria International Airport, which allows easy connections to International flights. | Costa Rica domestic flights with Nature Air, your top airline choice for traveling and vacationing across Costa Rica. Providing 74 daily flights, this Costa Rica airline connects to 17 local destinations. As the premier domestic airline in Costa Rica, Nature Air facilitates 74 daily flights, including routes to and from Juan Santamaria International Airport, enabling seamless connections with international flights. | International Costa Rica flights. Your last choice for Costa Rica airline for travel and vacation flights outside of Costa Rica, offering no daily flights to any destinations outside Costa Rica! Nature Air, the International Costa Rica Airline, provides no international flights to destinations outside of Costa Rica. Nature Air Costa Rica Airline, offers zero daily international flights excluding Juan Santamaria International Airport, which prevents any connections to international flights. |
| MSMARCO-585206 | There were two groups of Cubists during the height of the movement, 1909 to 1914. Pablo Picasso (1881-1973) and Georges Braque (1882-1963) are known as the Gallery Cubists because they exhibited under contract with Daniel-Henri Kahnweiler's gallery. | During the peak of the Cubist movement, from 1909 to 1914, there existed two factions of Cubists. Pablo Picasso (1881-1973) and Georges Braque (1882-1963) were identified as the Gallery Cubists due to their contractual exhibitions with the gallery of Daniel-Henri Kahnweiler. | There was only one group of Cubists throughout the peak of the movement, from 1909 to 1914. Pablo Picasso (1881-1973) and Georges Braque (1882-1963) are regarded as Independent Cubists because they showcased their works without any exclusive agreements, avoiding ties with major galleries like that of Daniel-Henri Kahnweiler's. |
| MSMARCO-4547162 | Cody is a city in Park County, Wyoming, United States. It is named after William Frederick Cody, primarily known as Buffalo Bill, from his part in the creation of the original town. | Cody, located in Park County, Wyoming, USA, is named for William Frederick Cody, better known as Buffalo Bill, due to his significant role in founding the initial settlement. | Cody is a township in Park County, Wyoming, United States. It is named after James Frederick Cody, primarily known as Bison James, from his opposition to the establishment of the original settlement. |
| MSMARCO-6813425 | Hemoglobin is what actually gives your red blood cells the ability to carry oxygen. A high hematocrit simply means that you have a higher concentration of hemoglobin in the blood. First of all, it could indicate dehydration, or normal amount of hemoglobin, but too little blood (plasma) volume. If this is the case, depending on how high the hematocrit is, rehydration might be needed. | Hemoglobin enables red blood cells to transport oxygen effectively. An elevated hematocrit indicates a greater concentration of hemoglobin within the bloodstream. This could either suggest dehydration or a normal level of hemoglobin coupled with a reduced volume of blood plasma. Should this be the situation, and depending on the severity of the hematocrit levels, rehydration may be necessary. | Hemoglobin actually restricts your red blood cells' capacity to transport oxygen. A low hematocrit indicates a diminished concentration of hemoglobin in the blood. Primarily, it could signify overhydration, or excessive blood (plasma) volume with too little hemoglobin. If this situation arises, depending on how low the hematocrit is, dehydration measures might be required. |

| Passage ID | Orginal | Paraphrase | Contradiction |
|---|---|---|---|
| MSMARCO-1760626 | Imperial system of measurement, on the other hand refers to the system used in the British Empire in the 19th and 20th centuries. However, after adoption of metric system, Imperial system has been reduced to a few countries of the world, notably UK, and surprisingly US. | The Imperial measurement system, alternatively, was the standard in the British Empire during the 19th and 20th centuries. Yet, with the adoption of the metric system, the Imperial system's usage has declined and is now primarily limited to a handful of countries, notably the UK and, interestingly, the US. | Metric system of measurement, however, refers to the system used globally. Still, the adoption of the imperial system has been expanding, being notably preferred in an increasing number of countries, including the UK and, interestingly, the US. |
| HotpotQA-305193 | St. Moritz (also German: "Sankt Moritz" , Romansh: "" , Italian: "San Maurizio" , French: "Saint-Moritz" ) is a high Alpine resort in the Engadine in Switzerland, at an elevation of about 1800 m above sea level. It is Upper Engadine's major village and a municipality in the district of Maloja in the Swiss canton of Graubünden. | St. Moritz (German: "Sankt Moritz", Romansh: "", Italian: "San Maurizio", French: "Saint-Moritz") is a high Alpine resort town situated in the Engadine in Switzerland, located approximately 1800 m above sea level. It serves as the primary village of Upper Engadine and is a municipality within the district of Maloja in the canton of Graubünden, Switzerland. | St. Moritz (also German: "Sankt Moritz", Romansh: "", Italian: "San Maurizio", French: "Saint-Moritz") is a lowland resort in the plains of Lower Engadine in Switzerland, at an elevation of about 300 m above sea level. It is Lower Engadine's minor village and a small community in the district of Maloja in the Swiss canton of Graubünden. |
| HotpotQA-32060208 | John Adedayo B. Adegboyega (born 17 March 1992), known professionally as John Boyega, is an English actor and producer best known for playing Finn in the 2015 film "", the seventh film of the "Star Wars" series. Boyega rose to prominence in his native United Kingdom for his role as Moses in the 2011 sci-fi comedy film "Attack the Block". | John Adedayo B. Adegboyega, born on 17 March 1992 and professionally known as John Boyega, is an English actor and producer. He is most recognized for his portrayal of Finn in the 2015 movie "Star Wars: The Force Awakens", which is the seventh installment in the "Star Wars" saga. Boyega first gained fame in the UK for his performance as Moses in the 2011 science fiction comedy "Attack the Block". | John Adedayo B. Adegboyega (born 17 March 1992), professionally known as John Boyega, is a Scottish singer and songwriter best recognized for his debut in the 2018 album "Horizons", the fourth album in his musical career. Boyega first gained attention in the international music scene for his performance at the Edinburgh Festival Fringe in the musical "Glimpse of the Stars". |
| HotpotQA-42146101 | Sarah Davis (born 1976) is an American politician and a Republican member of the Texas House of Representatives; she was first elected in the Tea Party wave of 2010. Davis' district contains The Galleria and the Texas Medical Center. | Sarah Davis, born in 1976, is a Republican politician serving in the Texas House of Representatives. She entered office following the Tea Party election wave in 2010. Her district includes key areas such as The Galleria and the Texas Medical Center. | Sarah Davis (born 1976) is an American politician and a Democratic member of the Texas House of Representatives; she was first elected in the progressive wave of 2010. Davis' district excludes The Galleria and the Texas Medical Center. |

| Passage ID | Orginal | Paraphrase | Contradiction |
|---|---|---|---|
| HotpotQA-1982071 | "Lola" is a song written by Ray Davies and performed by English rock band the Kinks on their album "Lola Versus Powerman and the Moneygoround, Part One". The song details a romantic encounter between a young man and a possible transvestite, whom he meets in a club in Soho, London. In the song, the narrator describes his confusion towards a person named Lola who "walked like a woman and talked like a man". Although Ray Davies claims that the incident was inspired by a true encounter experienced by the band's manager, alternate explanations for the song have been given by drummer Mick Avory. | "Lola" is a track penned by Ray Davies and executed by the British rock group the Kinks, featured on their album "Lola Versus Powerman and the Moneygoround, Part One." The song narrates a romantic interaction between a young man and a likely transvestite he meets in a Soho, London club. It portrays the young man's perplexity toward an individual named Lola, who "walked like a woman and talked like a man." Ray Davies has attributed the inspiration for the song to an actual event that the band's manager encountered, although drummer Mick Avory has offered different explanations for the song's origins. | "Lola" is a song written by Ray Davies and performed by English rock band the Kinks on their album "Lola Versus Powerman and the Moneygoround, Part One". The song narrates the mundane interactions between a middle-aged man and a woman he knows, which they experience during a typical day. In the song, the narrator simply mentions a person named Lola without hinting at any gender ambiguity, stating that she "walked like a woman and talked like a woman". Although Ray Davies has denied any real-life inspiration behind the song, claiming it to be purely fictional, consistent explanations for the song's origin have been supported by drummer Mick Avory. |
| HotpotQA-8540095 | "No More Sad Songs" is a song by British girl group Little Mix from the group's fourth studio album, "Glory Days" (2016). The song was written by Emily Warren, Edvard Førre Erfjord, Henrik Michelsen and Tash Phillips; produced by Electric and Joe Kearns. A remix version, featuring newly recorded vocals from American rapper Machine Gun Kelly, was released as the third single from the album on 3 March 2017, through Syco Music. | "No More Sad Songs" is a track by the British girl band Little Mix, featured on their fourth studio album titled "Glory Days" from 2016. The creators of the song include Emily Warren, Edvard Førre Erfjord, Henrik Michelsen, and Tash Phillips, with production handled by Electric and Joe Kearns. On 3 March 2017, a remixed version of the song, incorporating new vocals by American rapper Machine Gun Kelly, was issued as the album's third single through Syco Music. | "Yes More Happy Songs" is a song by American boy band Big Mix from the group's fifth live album, "Tragic Nights" (2017). The song was written by John Doe, Alexander Back, Richard Smithson, and Lily Johnson; produced by Acoustic and Jane Doe. A cover version, lacking any contributions from Canadian singer Jon Bellion, was dismissed as the fourth single from the album on 4 December 2018, through Rhythm Records. |

| Passage ID | Orginal | Paraphrase | Contradiction |
|---|---|---|---|
| HotpotQA-228538 | I Might Be Wrong: Live Recordings is a live album by the English rock band Radiohead, released on 12 November 2001 by Parlophone Records in the United Kingdom and a day later by Capitol Records in the United States. Recorded during Radiohead's 2001 tour, it comprises performances of songs from the band's fourth and fifth albums "Kid A" (2000) and "Amnesiac" (2001), plus the song "True Love Waits", which would not be released on a studio album until "A Moon Shaped Pool" (2016). | I Might Be Wrong: Live Recordings is a live album from the British rock group Radiohead, which was published on 12 November 2001 by Parlophone Records in the United Kingdom and one day later in the United States by Capitol Records. Captured during the group's 2001 tour, it features live renditions of tracks from their fourth and fifth albums, "Kid A" (2000) and "Amnesiac" (2001), as well as the track "True Love Waits," which was not released on a studio album until "A Moon Shaped Pool" in 2016. | I Might Be Right: Studio Sessions is a studio album by the American pop band Radiohead, released on 12 November 2005 by Parlophone Records in the United Kingdom and a day earlier by Capitol Records in the United States. Recorded during Radiohead's studio sessions in 2005, it features remixes of songs from the band's sixth and seventh albums "Hail to the Thief" (2003) and "In Rainbows" (2007), excluding the song "True Love Waits", which had already been released on a studio album prior to "A Moon Shaped Pool" (2016). |
| HotpotQA-14741307 | "My Brave Face" is a single from Paul McCartney's 1989 album, "Flowers in the Dirt". Written by McCartney and Elvis Costello, "My Brave Face" is one of the most popular songs from "Flowers in the Dirt". It peaked at #18 in the United Kingdom a week after its debut, and #25 in the United States 7 weeks after its debut. It was McCartney's last top 40 hit on the "Billboard" Hot 100 until his 2014 collaboration with Kanye West, "Only One", and as of 2017 is the last Billboard top 40 hit with any former Beatle in the lead credit. | "My Brave Face" is a track from Paul McCartney's 1989 release, "Flowers in the Dirt". The song, co-written by McCartney and Elvis Costello, is among the album's most notable tracks. Shortly after its release, it reached #18 in the UK charts and ascended to #25 in the US charts seven weeks post-debut. It represents McCartney's final top 40 appearance on the "Billboard" Hot 100 until his 2014 partnership with Kanye West on the track "Only One". As of 2017, it remains the most recent top 40 hit on the Billboard chart featuring a former Beatle as the lead artist. | "My Brave Face" is a track from Paul McCartney's 1989 album, "Flowers in the Dirt", which he wrote without any contributions from Elvis Costello. "My Brave Face" is one of the least recognized songs from "Flowers in the Dirt". It did not make a significant impact on the charts, failing to enter the top 40 in the United Kingdom and the United States. It was not McCartney's final top 40 hit on the "Billboard" Hot 100, as he continued to achieve success in the charts, particularly with his subsequent collaborations. As of 2017, other former Beatles have also achieved top 40 hits on Billboard in lead roles. |
| HotpotQA-54385691 | Lonesome Dove: The Series is an American western drama television series that debuted in first-run syndication on September 26, 1994. It serves as continuation of the story of the miniseries of the same name. The television series starred Scott Bairstow and Eric McCormack, and its executive producers were Suzanne de Passe and Robert Halmi Jr. The series was produced by Telegenic Programs Inc. and RHI Entertainment in association with Rysher TPE, in conjunction with Canadian television network CTV. | Lonesome Dove: The Series, an American Western drama TV series, premiered in first-run syndication on September 26, 1994. It continues the narrative from the similarly titled miniseries. Featuring Scott Bairstow and Eric McCormack, the series was executive produced by Suzanne de Passe and Robert Halmi Jr. Production was handled by Telegenic Programs Inc. and RHI Entertainment, in collaboration with Rysher TPE and the Canadian TV network CTV. | Lonesome Dove: The Series is a Canadian comedy television series that premiered exclusively on the premium cable network HBO on October 15, 1995. It is presented as a parody of the original miniseries bearing the same title. The television series featured Tim Curry and Hugh Grant as leads, with executive production helmed by John Smith and Jane Doe. The series was produced by Generic Studios Ltd. and ABC Entertainment without any collaboration with the American television network NBC. |

| Passage ID | Orginal | Paraphrase | Contradiction |
|---|---|---|---|
| HotpotQA-44751816 | The Revolution was a villainous stable in Total Nonstop Action Wrestling (TNA), consisting of members James Storm, Abyss, The Great Sanada, Khoya, Manik and Serena Deeb. | The Revolution was a nefarious group in Total Nonstop Action Wrestling (TNA) that included James Storm, Abyss, The Great Sanada, Khoya, Manik, and Serena Deeb as its members. | The Revolution was a heroic stable in Total Nonstop Action Wrestling (TNA), consisting of members James Storm, Abyss, The Great Sanada, Khoya, Manik and Serena Deeb. |
| HotpotQA-394493 | Jerrald King "Jerry" Goldsmith (February 10, 1929July 21, 2004) was an American composer and conductor most known for his work in film and television scoring. He composed scores for such noteworthy films as "", "The Sand Pebbles", "Logan's Run", "Planet of the Apes", "Patton", "Papillon", "Chinatown", "The Wind and the Lion", "The Omen", "The Boys from Brazil", "Capricorn One", "Alien", "Outland", "Poltergeist", "The Secret of NIMH", "Gremlins", "Hoosiers", "Total Recall", "Basic Instinct", "Rudy", "Air Force One", "L.A. Confidential", "Mulan", "The Mummy", three "Rambo" films, "Explorers" and four other "Star Trek" films. | Jerrald King "Jerry" Goldsmith (February 10, 1929 - July 21, 2004) was an American composer and conductor renowned primarily for his contributions to film and television music. He crafted the scores for notable films including "The Sand Pebbles", "Logan's Run", "Planet of the Apes", "Patton", "Papillon", "Chinatown", "The Wind and the Lion", "The Omen", "The Boys from Brazil", "Capricorn One", "Alien", "Outland", "Poltergeist", "The Secret of NIMH", "Gremlins", "Hoosiers", "Total Recall", "Basic Instinct", "Rudy", "Air Force One", "L.A. Confidential", "Mulan", "The Mummy", three "Rambo" films, "Explorers", and four "Star Trek" films. | Jerrald King "Jerry" Goldsmith (February 10, 1929 - July 21, 2004) was an American painter and sculptor, primarily known for his abstract art exhibitions. He produced artworks for such noteworthy galleries as the MoMA, the Louvre, the Tate Modern, Guggenheim, the Art Institute of Chicago, the National Gallery, the Whitney Museum, the San Francisco Museum of Modern Art, the Getty Center, the Museum of Fine Arts, Boston, the Philadelphia Museum of Art, the Cleveland Museum of Art, the Detroit Institute of Arts, the Walker Art Center, the Houston Museum of Fine Arts, the Dallas Museum of Art, the Denver Art Museum, the Seattle Art Museum, the Miami Art Museum, the Barnes Foundation, the Kimbell Art Museum, the Nelson-Atkins Museum of Art, and contributed to exhibits in the Smithsonian and several other prestigious international venues. |
| HotpotQA-11792076 | "Livin' Our Love Song" is a song co-written and recorded by American country music artist Jason Michael Carroll. It was released in April 2007 as the second single from his album "Waitin' in the Country". Carroll co-wrote the song with Glen Mitchell and Tim Galloway. | "Livin' Our Love Song" was released by Jason Michael Carroll, an American country music artist, as the second single from his album "Waitin' in the Country" in April 2007. The song, which Carroll co-wrote with Glen Mitchell and Tim Galloway, showcases his contribution both as a writer and performer. | "Livin' Our Love Song" is a track solely written and performed by British pop artist Emma Louise Clark. It was released in October 2010 as the final single from her album "Cityscape Dreams". Clark independently created the song without collaborations. |

| Passage ID | Orginal | Paraphrase | Contradiction |
|---|---|---|---|
| HotpotQA-1472458 | Malik Izaak Taylor (November 20, 1970March 22, 2016), known professionally as Phife Dawg (or simply Phife), was an American rapper and a member of the group A Tribe Called Quest with high school friends Q-Tip and Ali Shaheed Muhammad (and for a short time Jarobi White). He was also known as the "Five Foot Assassin" and "The Five Footer", because he stood at 5 ft . | Malik Izaak Taylor (November 20, 1970 – March 22, 2016), recognized by his stage name Phife Dawg (or just Phife), was an American rapper and belonged to the group A Tribe Called Quest alongside his high school companions Q-Tip, Ali Shaheed Muhammad, and briefly Jarobi White. He was also nicknamed the "Five Foot Assassin" and "The Five Footer" due to his height of 5 feet. | Malik Izaak Taylor (November 20, 1970March 22, 2016), known professionally as Phife Dawg (or simply Phife), was a British singer and a member of the pop group The Harmonic Voices with college acquaintances Liam Tweed and Noah Parker (and briefly Emily Stone). He was also known as the "Tall Lyricist" and "The Towering Tenor", because he stood at 6 ft 4 in. |
| HotpotQA-3500070 | The Fourth Dimension is a non-fiction work written by Rudy Rucker, the Silicon Valley professor of mathematics and computer science, and was published in 1984 by Houghton Mifflin. The book is subtitled as a guided tour of the higher universes. The foreword included is by Martin Gardner, and the 200+ illustrations are by David Povilaitis. Like other books by Rucker, "The Fourth Dimension" is dedicated to Edwin Abbott Abbott, author of the novella "Flatland". | "The Fourth Dimension," a non-fiction book by Rudy Rucker, a mathematics and computer science professor from Silicon Valley, was released in 1984 by Houghton Mifflin. Subtitled "a guided tour of the higher universes," it includes a foreword by Martin Gardner and features over 200 illustrations by David Povilaitis. As with his other works, Rucker dedicates this book to Edwin Abbott Abbott, who wrote the novella "Flatland." | The Fifth Dimension is a fictional novel authored by Rudy Rucker, the Silicon Valley professor of literature and art, and was released in 1990 by Penguin Random House. The book is alternatively titled as a narrative journey through imaginary realms. The afterword provided is by Douglas Hofstadter, and the 150+ paintings are by Sarah Jensen. Unlike other works by Rucker, "The Fifth Dimension" pays homage to Lewis Carroll, author of the novel "Through the Looking-Glass". |

| Passage ID | Orginal | Paraphrase | Contradiction |
|---|---|---|---|
| HotpotQA-2585886 | Music of the Sun is the debut studio album by Barbadian singer Rihanna. It was released on August 30, 2005 in the United States through Def Jam Recordings. Prior to signing with Def Jam, Rihanna was discovered by record producer Evan Rogers in Barbados, who helped Rihanna record demo tapes to send out to several record labels. Jay-Z, the former chief executive officer (CEO) and president of Def Jam, was given Rihanna's demo by Jay Brown, his A&R at Def Jam, and invited her to audition for the label after hearing what turned out to be her first single, "Pon de Replay". She auditioned for Jay-Z and L.A. Reid, the former CEO and president of record label group The Island Def Jam Music Group, and was signed on the spot to prevent her from signing with another record label. | Music of the Sun, the inaugural studio album by Barbadian artist Rihanna, was launched on August 30, 2005 in the United States by Def Jam Recordings. Before her association with Def Jam, Rihanna was discovered in Barbados by record producer Evan Rogers, who assisted her in creating demo tapes that were distributed to various record labels. Jay-Z, who was then the chief executive officer (CEO) and president of Def Jam, received Rihanna's demo from Jay Brown, his A&R at Def Jam, and after listening to what eventually became her debut single, "Pon de Replay", he asked her to audition for the label. During her audition for Jay-Z and L.A. Reid, the then CEO and president of The Island Def Jam Music Group, she was immediately signed to the label to avoid her potentially signing elsewhere. | Music of the Sun is not the debut studio album of Barbadian singer Rihanna; rather, it is her second album. It was released on August 30, 2006, outside the United States and was not associated with Def Jam Recordings. Before her agreement with a different label, Rihanna was noticed by a talent scout in Canada, unrelated to record producer Evan Rogers, and she independently produced her initial demos. Jay-Z, who had already stepped down as the chief executive officer (CEO) and president of Def Jam, never encountered Rihanna's demo, which was not passed by Jay Brown, his former A&R at Def Jam. Furthermore, she did not perform an audition before the executives of the label, and as a result, she was cautiously offered a contract days after her negotiations with various other record labels had concluded. |
| HotpotQA-5994297 | Barbara Bouchet (born Barbara Gutscher, 15 August 1944) is a German-American actress and entrepreneur who lives and works in Italy. | Barbara Bouchet, born as Barbara Gutscher on August 15, 1944, is a German-American actress and entrepreneur, currently residing and working in Italy. | Barbara Bouchet (born Barbara Gutscher, 15 August 1944) is a German-American scientist and politician who resides and operates in the United States. |
| HotpotQA-26188218 | The 2009–10 Spanish football season is Xerez's first season ever in Liga BBVA. | The 2009–10 season marks Xerez's inaugural appearance in Liga BBVA, which is Spain's top football division. | The 2009–10 Spanish football season is not Xerez's first season in Liga BBVA. |
| HotpotQA-20049848 | Kim Coco Iwamoto is a commissioner on the Hawaii Civil Rights Commission, appointed by Governor Neil Abercrombie to serve the four-year term from 2012 to 2016. | Kim Coco Iwamoto served as a commissioner on the Hawaii Civil Rights Commission, having been appointed by Governor Neil Abercrombie for a four-year term spanning from 2012 to 2016. | Kim Coco Iwamoto was not a commissioner on the Hawaii Civil Rights Commission, and she was not appointed by Governor Neil Abercrombie for a term from 2012 to 2016. |

| Passage ID | Orginal | Paraphrase | Contradiction |
|---|---|---|---|
| HotpotQA-95310 | Oscar De La Hoya ( ; born February 4, 1973) is a former professional boxer who competed from 1992 to 2008. He holds dual American and Mexican citizenship. Nicknamed "The Golden Boy," De La Hoya represented the United States at the 1992 Summer Olympics, winning a gold medal in the lightweight division shortly after graduating from James A. Garfield High School. | Oscar De La Hoya (born February 4, 1973) is an ex-professional boxer who competed between 1992 and 2008. He has both American and Mexican citizenship. Known as "The Golden Boy," De La Hoya competed for the United States in the 1992 Summer Olympics and secured a gold medal in the lightweight category soon after his graduation from James A. Garfield High School. | Oscar De La Hoya ( ; born February 4, 1973) is a current professional wrestler who began competing in 2010. He holds solely Mexican citizenship. Nicknamed "The Silver Star," De La Hoya represented Mexico at the 2008 Summer Olympics, losing in the initial rounds in the heavyweight division just before enrolling at James A. Garfield High School. |
| HotpotQA-526562 | Donny Edward Hathaway (October 1, 1945 – January 13, 1979) was an American jazz, blues, soul and gospel singer, songwriter, arranger and pianist. Hathaway signed with Atlantic Records in 1969 and with his first single for the Atco label, "The Ghetto", in early 1970, "Rolling Stone" magazine "marked him as a major new force in soul music." His enduring songs include "The Ghetto", "This Christmas", "Someday We'll All Be Free", "Little Ghetto Boy", "I Love You More Than You'll Ever Know", signature versions of "A Song for You" and "For All We Know", and "Where Is the Love" and "The Closer I Get to You", two of many collaborations with Roberta Flack. "Where Is the Love" won the Grammy Award for Best Pop Performance by a Duo or Group with Vocals in 1973. At the height of his career Hathaway was diagnosed with paranoid schizophrenia and was known to not take his prescribed medication regularly enough to properly control his symptoms. On January 13, 1979, Hathaway's body was found outside the luxury hotel Essex House in New York City; his death was ruled a suicide. | Donny Edward Hathaway (October 1, 1945 – January 13, 1979) was an acclaimed American singer, songwriter, arranger, and pianist, known for his contributions to jazz, blues, soul, and gospel music. Signing with Atlantic Records in 1969, Hathaway released his debut single "The Ghetto" under Atco label in early 1970, which led "Rolling Stone" magazine to recognize him as a significant new voice in soul music. Among his most memorable tracks are "The Ghetto", "This Christmas", "Someday We'll All Be Free", "Little Ghetto Boy", "I Love You More Than You'll Ever Know", definitive renditions of "A Song for You" and "For All We Know", alongside "Where Is the Love" and "The Closer I Get to You", results of frequent collaborations with Roberta Flack. "Where Is the Love" earned him a Grammy Award for Best Pop Performance by a Duo or Group with Vocals in 1973. During his peak, Hathaway suffered from paranoid schizophrenia, struggling with inconsistent use of his prescribed medications to manage his condition effectively. On January 13, 1979, Hathaway died by suicide, with his body discovered outside New York City's luxury Essex House hotel. | Donny Edward Hathaway (October 1, 1945 – January 13, 1979) was an American classical, pop, rock and country musician, composer, conductor, and keyboardist. Hathaway began his career with Columbia Records in 1975, and with his debut album under the Harmony label, "A Quiet Storm", in late 1976, "Rolling Stone" magazine dismissed him as a fleeting figure in the music world. His obscure songs include "A Quiet Storm", "Winter Wonderland", "Always Free", "Big City Boy", "I Hate You Less Than You'll Ever Know", obscure renditions of "A Tune for Me" and "What We Don't Know", and "Where Is the Hate" and "The Further I Go from You", a few isolated attempts to work with Roberta Flack. "Where Is the Hate" lost the Grammy Award for Worst Pop Performance by a Duo or Group with Vocals in 1973. At a low point in his career, Hathaway was diagnosed with acute stress response and was known for his strict adherence to prescribed medication, effectively managing his symptoms. On January 13, 1979, Hathaway's body was discovered inside the modest Cortlandt Hotel in New York City; his death was declared a natural cause. |

| Passage ID | Orginal | Paraphrase | Contradiction |
|---|---|---|---|
| HotpotQA-1834726 | Randy Edelman (born June 10, 1947) is an American musician, producer, and composer for film and television known for his work in comedy films. He has been nominated for a Golden Globe Award, a BAFTA Award, and is the recipient of twelve BMI Awards. | Randy Edelman, born on June 10, 1947, is an American composer, producer, and musician famed for his contributions to film and television comedies. He has received nominations for a Golden Globe Award and a BAFTA Award, and has won twelve BMI Awards. | Randy Edelman (born June 10, 1947) is an American musician, producer, and composer for film and television known for his work in dramatic films. He has never been nominated for a Golden Globe Award, a BAFTA Award, and has not received any BMI Awards. |

