# OpenReview forum: "Contradiction Retrieval via Contrastive Learning with Sparsity"
_ICML.cc/2025/Conference — ICML 2025 poster_

### Official Review · Reviewer_6o5M · 2025-03-15

**Overall Recommendation:** 3

**Summary:**

The paper proposes a new method named SparseCL to leverage specially trained sentence embeddings designed to preserve subtle, contradictory nuances between sentences. SparseCL utilizes a combined metric of cosine similarity and a sparsity function to efficiently identify and retrieve documents that contradict a given query. Experiments are performed on the Arguana, MSMARCO, and HotpotQA datasets to demonstrate the effectiveness of the proposed method. The authors apply the proposed contradiction retrieval method to two downstream settings including corpus cleaning and natural language inference to highlight the practical benefits of the proposed approach in real-world scenarios.

**Claims And Evidence:**

Yes. The claims are mostly well-supported by the experimental results.

**Essential References Not Discussed:**

No. The references are well-cited and discussed.

**Experimental Designs Or Analyses:**

Yes. The experimental designs are overall reasonably designed.

**Methods And Evaluation Criteria:**

Yes. The proposed method can improve the baseline without introducing much burden.

**Other Comments Or Suggestions:**

- Figure 1 is not referenced.
- The caption of Figure 2 L196 and L199: left/right should be upper/bottom?

**Other Strengths And Weaknesses:**

Strengths:
- The paper is overall well-written and organized.
- Besides the cosine similarity, the paper proposes to use Hoyer sparsity as the metric for contradiction retrieval without complex pair-wise computation for reranking.
- The experimental validation is performed on three public datasets and demonstrates the effectiveness of the proposed method.

Weakness:
- The overall design is rather simple with a well-known sparsity metric.
- Besides the provided baselines (BGE, UAE, GTE), does the proposed method perform well on an LLM-based encoding method?

**Questions For Authors:**

N/A

**Relation To Broader Scientific Literature:**

The proposed contradiction retrieval method may benefit areas like corpus cleaning and natural language inference.

**Theoretical Claims:**

Yes. Theoretical Claims are well supported.

---

> ### Author Rebuttal · Authors · 2025-04-01
>
> We greatly appreciate your insightful comments provided for our work. Here are our responses to the proposed weaknesses and questions.
>
> **W1: The overall design is rather simple with a well-known sparsity metric.**
>
> As shown in Table 4, the idea of SparseCL is not limited to Hoyer sparsity, but also works on many other sparsity functions. We believe that simple idea with superior performance is actually an advantage.
>
> **W2: Besides the provided baselines (BGE, UAE, GTE), does the proposed method perform well on an LLM-based encoding method?**
>
> Thank you for pointing out this consideration! As of September 2024, the 'gte-large-v1.5' model, achieving an NDCG score of 72, performed best on the Arguana dataset among models with fewer than 1 billion parameters. Additionally, LLM-based embeddings were not effective on Arguana during that time; for instance, SFR-Embedding-Mistral scored 67 in NDCG, and Voyage-lite-02-instruct scored 70. We believe that the three models we tested, varying in size, backbone, and performance, sufficiently demonstrate the effectiveness of our SparseCL method.
>
> **Typo**
>
> Thank you for correcting the typos. We will fix them in the final version.

---

### Official Review · Reviewer_isYx · 2025-03-22

**Overall Recommendation:** 4

**Summary:**

This paper addresses the task of contradiction retrieval—retrieving documents or passages that explicitly refute a given query. The authors introduce SPARSECL, a method that augments standard contrastive learning for sentence embeddings with a sparsity measure (specifically, the Hoyer measure) to capture subtle contradictory nuances. The proposed approach combines traditional cosine similarity with a sparsity-based score to more effectively identify contradictions. Extensive experiments are performed on the Arguana dataset as well as on synthetic contradiction datasets constructed from MSMARCO and HotpotQA, with additional applications in corpus cleaning and natural language inference.

**Claims And Evidence:**

Key Claims:
1. SPARSECL can enhance contradiction retrieval over standard similarity-based methods.
2. Incorporating the Hoyer sparsity measure with cosine similarity yields measurable improvements (e.g., in NDCG@10) across multiple datasets.
3. The method generalizes across datasets and has practical downstream applications (e.g., corpus cleaning).

Evidence:
1. The paper presents quantitative improvements (e.g., up to 11.0% average gain in retrieval metrics) compared to baselines such as standard contrastive learning and prompt-augmented approaches.
2. Ablation studies and additional zero-shot tests are provided, reinforcing the claim that the sparsity measure contributes uniquely to capturing contradictions.

**Essential References Not Discussed:**

While the paper cites relevant work on counter-argument retrieval (e.g., Shi et al. (2023) on the Bipolar-encoder), it might benefit from referencing additional recent studies on robust retrieval or adversarial defenses in retrieval systems, which could provide context regarding alternative approaches to handling contradictory information.

A discussion on recent developments in fact-checking and misinformation detection could also broaden the context of its contributions.

**Experimental Designs Or Analyses:**

Design Strengths:
Ablation studies dissect the contribution of each component (e.g., effect of adding paraphrases, alternative sparsity functions). Downstream applications (such as corpus cleaning) are tested, demonstrating the method’s utility beyond standard retrieval benchmarks.

Potential Issues:
Some experiments rely on synthetic data generated by GPT-4, which might not fully capture the complexities of naturally occurring contradictions. The evaluation on the Arguana dataset suggests that in some cases, even similarity-based methods perform reasonably well, potentially masking the unique contribution of the sparsity component.

**Methods And Evaluation Criteria:**

The authors fine-tune pre-trained sentence embedding models via a modified contrastive learning loss that rewards high Hoyer sparsity for contradiction pairs while using similar (non-contradictory) pairs as negatives. A combined scoring function—cosine similarity plus an α-weighted Hoyer score—is used at test time to rank candidate passages.

Retrieval performance is measured using NDCG@10 on Arguana, as well as on modified MSMARCO and HotpotQA datasets. Additional evaluations include ablation studies, zero-shot generalization tests, and experiments on corpus cleaning.

**Other Comments Or Suggestions:**

Failure modes: Discuss potential limitations or failure cases in more detail, particularly concerning the generalization to varied contradiction types.

**Other Strengths And Weaknesses:**

Strengths:
1. Novel use of a sparsity measure (Hoyer) to capture contradiction nuances.
2. Comprehensive experimental validation across multiple datasets and applications.
3. Clear exposition of the limitations of standard cosine-based retrieval and the motivation for a new metric.

Weaknesses:
1. Dependence on synthetic data for some evaluations might limit external validity.
2. The sensitivity to the hyperparameter α and potential computational implications in high-dimensional spaces are not fully explored.
3. Some parts of the theoretical analysis could be more tightly integrated with empirical findings.

**Questions For Authors:**

1. Hyperparameter Sensitivity: How sensitive is the retrieval performance to the choice of the α parameter? Could you provide a more detailed analysis of its tuning across different datasets?

2. Data Generation Concerns: Given that some datasets rely on GPT-4–generated contradictions, how do you expect SPARSECL to perform on datasets containing naturally occurring contradictions?

3. High-Dimensional Effects: Can you elaborate on any potential issues that might arise from applying the Hoyer sparsity measure in high-dimensional embedding spaces, and whether alternative sparsity measures might offer complementary advantages?

**Relation To Broader Scientific Literature:**

The paper is well-situated within the literature on contrastive learning and sentence embeddings, and it draws appropriate connections to fact verification and counter-argument retrieval. Its focus on non-similarity-based retrieval differentiates it from prior work, although a deeper discussion comparing its approach with alternative methods for detecting contradictions (e.g., adversarial training methods) would further clarify its novelty.

**Theoretical Claims:**

The paper argues that cosine similarity’s transitivity limits its ability to capture contradiction (since similar sentences may both contradict a third without contradicting each other).

In contrast, the Hoyer sparsity measure is shown (via propositions in the Appendix) to be non-transitive, allowing it to better represent the localized differences that signify contradiction.

The provided proofs (Propositions C.1 and C.2) appear sound under the stated assumptions, but the practical implications in high-dimensional settings might warrant further discussion.

---

> ### Author Rebuttal · Authors · 2025-04-01
>
> We greatly appreciate your insightful comments on our work. Here are our responses to the proposed weaknesses and questions.
> W1 & W2 see below Q2 and Q1
> **W3: Some parts of the theoretical analysis could be more tightly integrated with empirical findings.**
>
> We are not claiming that we have theoretical analysis for characterizing the contradiction relationship. The two propositions in Appendix serve as motivating examples to show the fundamental limitation of the cosine metric in representing contradictions and how the sparsity metric can bypass that.
>
> **Q1: hyperparameter sensitivity**
>
> Here is the way we tune the $\alpha$ parameter: From the range [0, 10], we first divide the range into 10 intervals, calculating the NDCG@10 score on the validation set for each interval’s midpoint, and then diving into that interval for a finer search. We stop when the interval range is smaller than 0.01. We will collect all the $\alpha$ values and investigate their sensitivity in the final version.
>
> **Q2: data generation concerns**
>
> Please see our experimental results on Arguana, NLI datasets, where contradictions are collected from natural human-written sources.
>
> **Q3: High-Dimensional Effects**
>
> We believe that the current embeddings are already in a high-dimensional space (e.g. bge-base has dimensionality 768) and our experiments work well there. Please see our ablation study in Table 4 on different sparsity functions. All of them yield some improvement compared with the baseline, while our Hoyer sparsity yields the best performance. We believe that’s because simple sparsity functions have a more benign optimization landscape and thus are easier for models to optimize, which is worth future study.

---

### Official Review · Reviewer_gpUP · 2025-03-23

**Overall Recommendation:** 1

**Summary:**

This paper introduces SparseCL, a novel approach for contradiction retrieval using sparse-aware sentence embeddings.  The method addresses limitations of traditional similarity search and cross-encoder methods by training sentence embeddings to preserve sparsity of differences between embeddings of contradicting sentences. SparseCL utilizes a combined metric of cosine similarity and Hoyer sparsity to score and retrieve documents that contradict a query, aiming for efficiency and improved contradiction detection.

## update after rebuttal

Responses did not really address the methodological questions I had (which is to be expected, since the issues go to the heart of the experiments presented)

It seems other reviewers didn't find the dataset *HotPotQA-synthetic-rephrase* being labelled *HotPotQA* throughout as much of a problem as I did.  Clearly Figure 2 is impressive, though I feel that this also points to a separation between matching/contradicting passages that is too-good-to-be-true (and hence my skepticism about the datasets overall).

Rating remains "1: Reject"

**Claims And Evidence:**

The embedding training separation shown in Figure 2 is very impressive - the training is clearly effective according to the metrics.

In L271, the authors state "This pattern of enhancement was consistently observed regardless of whether the embedding models were fine-tuned or not."  This is a very puzzling statement, which prompted digging deeper.

Looking at the samples provided in Appendix H: It seems like many of the LLM-generated contradiction samples are 'simple contradictions' - including "no" and "not" far more than the straight paraphrases.  Which in turn leads to a reconsideration of what the MSMARCO and HotpotQA dataset results actually represent.  A plausible explanation for the experimental results is simply that the paraphrasing (positive or contradictory) preserves L2 similarity, while the sparse contradiction elements are just learning to recognise the signs of LLM-generated-contradiction.  This seems like a fundamental experimental flaw.

The fact that SparseCL *also* shows improvements on the Arguana dataset, which is a human-curated dataset, provides some counter-evidence to this artifact-only argument.  However, Arguana is also focused on counter-argument detection within debates, which might still exhibit specific textual patterns different from more general contradiction.  Moreover, the Arguana results show less dramatic performance - which is what raised the artifact questions above in the first place.

**Essential References Not Discussed:**

Nothing specific.

**Experimental Designs Or Analyses:**

As noted in 'Claims and Evidence', the MSMARCO and HotpotQA experimental design and results are overshadowed by the synthetic data offering 'short-cuts' to detection.

**Methods And Evaluation Criteria:**

The proposed SparseCL method, combining cosine similarity with a Hoyer sparsity measure on contrastively learned embeddings, is a sensible approach to address the limitations of purely similarity-based methods for contradiction retrieval. The use of contrastive learning with reversed positive/negative examples (contradictions as positive, paraphrases as negative) is also well-motivated for learning contradiction-aware embeddings.

**Other Comments Or Suggestions:**

### Typos / Fixes
* L043: Fix citation double bracketing
* L009: "while as far as we know" ... seems like the authors should be clear on this point
* L011: "non-simlarity" -> "non-similarity"
* L100: "In specific, they" -> "Specifically, they"
* L102: "This phenomenon bring our attention" -> "This phenomenon brought our attention"
* L105: "papers are different" -> "papers was different" (tense+number)
* L148: Fix ",."
* L157: "a score between [0, 1]" -> "a score in (0, 1)"
* L134: "so the authors of (Wachsmuth et al 2018) designed" -> "so Wachsmuth et al (2018) proposed"
* L154: Fix citation double bracketing
* L155: "fine-tune any pretrained" -> "fine-tune a pretrained"
* L421: Fix bracketing "selected from (Hurley...)" -> "selected from Hurley ... ()"
* L655: "(Shi et al., 2023) proposes" -> "Shi et al. (2023) proposes"

* Appendix C. "Two examples demonstrating the ..." -> "The ..." (these are proofs, not examples)

**Other Strengths And Weaknesses:**

### Strengths:

- **Novel Approach:** The SparseCL method is a novel and well-motivated approach for contradiction retrieval, addressing a gap in existing methods.
- **Clear Presentation:** The paper is generally well-written and clearly presents the method, experiments, and results.
- **Efficiency:** The method offers a computationally efficient alternative to cross-encoders.

### Weaknesses:

- **Synthetic Dataset Reliance:**  The heavy reliance on synthetically generated datasets (MSMARCO, HotpotQA) for evaluation is a significant weakness regarding generalizability and the potential for artifact exploitation.
- **Marginal Improvement on Arguana:** While improvements are shown on Arguana, they are less dramatic than on synthetic datasets, which could be interpreted as supporting the artifact-exploitation concern.
- **Need for Deeper Analysis:**  The paper could benefit from a deeper analysis of *why* SparseCL works, particularly on the synthetic datasets, and more detailed error analysis.

**Questions For Authors:**

### Synthetic Data Artifacts (Critical Question):

The MSMARCO and HotpotQA datasets are synthetically generated.  One concern is that SparseCL might be exploiting artifacts from the LLM's negative paraphrasing rather than capturing generalizable contradiction.  Could you please elaborate on how you have considered or mitigated this potential issue?  Specifically:
1) Have you analyzed failure cases or error types to see if SparseCL is overly sensitive to specific LLM-generated patterns?
1) Could you discuss the generalizability of your findings to real-world contradiction scenarios, given the reliance on synthetic data?
1) Are there any experiments or analyses you could perform to further validate that SparseCL captures genuine contradiction beyond LLM-specific artifacts?  For instance, could you compare performance on a subset of the synthetic data that is *manually* verified for contradiction quality, or explore datasets of contradictions not generated by LLMs?

*If your response clearly shows that I am mistaken about the potential issues with identifiable LLM 'smell' for the synthetic contradictions, then I would be happy to update my score*

**Relation To Broader Scientific Literature:**

The paper is well-positioned within the broader scientific literature on information retrieval and semantic representation learning. It correctly identifies the limitations of traditional similarity search for non-similarity-based tasks like contradiction retrieval.

The novelty lies in the specific combination of contrastive learning with a sparsity-inducing objective, tailored for contradiction detection, and the use of Hoyer sparsity.  The paper appropriately cites relevant works in these areas.

**Theoretical Claims:**

The paper includes a theoretical analysis in Appendix C about the non-transitivity of Hoyer sparsity and the transitivity of cosine similarity.  I have not rigorously checked the correctness of the proofs in Appendix C, but the intuition presented and the examples seem reasonable.  The theoretical motivation for using Hoyer sparsity to circumvent the transitivity of cosine similarity is clearly presented and makes sense.

---

> ### Author Rebuttal · Authors · 2025-04-01
>
> We greatly appreciate your insightful comments provided for our work. Here are our responses to the proposed weaknesses and questions.
>
> **L271 statement:**
> This means that our SparseCL method can be combined with either a zero-shot embedding model or a fine-tuned embedding model, and we demonstrate that in both cases, the retrieval performance with SparseCL surpasses the performance without SparseCL. We will clarify this in our final version.
>
> **Samples provided in Appendix H:**
> Thanks for pointing out this observation. We did a simple check that among the 40 examples we listed in Appendix H, 8 of the original passages have the string “no/not/non” and 17 of the generated contradictions have the string “no/not/non”. It is true that in some cases, contradictions are formed by adding negation words, but it is also worth noticing that in more than half of the cases, contradictions are judged in other non-trivial ways like opposite words or phrase replacement. We believe this represents the essence of a well-defined contradiction rather than a flaw.
>
> Second, it is still unclear how similarity-based retrieval methods can solve this task even if a certain style exists for generated contradictions. As we are looking for the opposite side of the query passage, rather than passages sharing the same style.
>
> **Concern about Arguana:**
> We agree that textual patterns exist for contradictions from any specific area. The reason that our method shows less dramatic performance in Arguana is that there aren't enough paraphrases within the corpus to confound the similarity-based retrieval method, because then it is likely that the counterargument is still the most similar passage within the corpus. In our ablation study (agreed by Reviewer RTuC), we manually add more paraphrases to the Arguana corpus, and we then observe a significant performance drop for the similarity-based retrieval method, and our method’s performance is unaffected. Please refer to Table 6.
>
> **W1: Synthetic Dataset Reliance:**
> We want to emphasize that, in addition to Arguana, our experiments on NLI tasks and data cleaning are conducted on real datasets. Please refer to Sections 4.4 and 4.5. For our data cleaning experiment, although the corrupted passages are generated by an LLM, they are similar enough to confuse current embedding models and cause a performance drop.
>
> **W2: Marginal Improvement on Arguana:**
> See above (Concern about Arguana)
>
> **W3: Need for Deeper Analysis:**
> Please refer to our ablation study, including sparsity function variants, data augmentation, and a case study in Appendix E to show its superiority against traditional cosine-based contrastive learning on the contradiction retrieval problem.
>
> **Typo:**
> We will fix all typos and citations for the camera ready version.
>
> **Q1 Synthetic Data Artifacts (Critical Question):**
> We have put in the prompt to encourage the generated content to be indistinguishable “in terms of length, language style, and format”. From the examples in Appendix H, we have manually checked and didn’t find any obvious language style difference between the paraphrases and contradictions. Given that it is hard to “prove” artifacts exist or not in generated passages, a manual check is the best we can do. We agree that certain key words like “no/not/non” may be used more often in contradictions, but this is the common practice to form contradictions, which also exist in human generated contradictions like Arguana.
>
> **Q2: the generalizability of your findings to real-world contradiction scenarios, given the reliance on synthetic data:**
> The motivation for us to generate contradiction pairs using LLM is the lack of natural contradiction pairs in real-world datasets besides Arguana and NLI datasets.
>
> **Q3: further validate that SparseCL captures genuine contradiction beyond LLM-specific artifacts:**
> We have sampled 500 passages from LLM generated contradictions and 2 people annotated the data to check the quality. Please refer to Appendix H. Therefore, we believe that the experiments run on our synthetic datasets are fair.

---

> > ### Comment · Reviewer_gpUP · 2025-04-04
> >
> > So, I don't see that your rebuttal has addressed my main concern : The MSMARCO and HotPotQA datasets referred to throughout the paper should all be properly identified as *MSMARCO-synthetic-rephrase* and *HotPotQA-synthetic-rephrase* - these were your actual train/val/test sets.  While "2 people annotated the data to check the quality" for these datasets, it could still easily be that an LLM could discover some kind of 'contradiction indicator' (my simple observation of 'no/not/non' was just the first one that leapt off the pages of your Appendix H).  So, I remain suspicious of the results in Sections 4.2 and 4.3.
> >
> > Similarly, Section 4.4 (Corpus Cleaning) could suffer from the same identification of short-cuts.
> >
> > The Natural Language Inference experiments of Section 4.5 are not based on the synthetic data that seems problematic to me.  However, the results in Table 5 are puzzling:
> >
> > * I see no reason for the SparseCL lines to be bolded.  Does having a lower score in each column have any significance?
> > * For an effective SparseCL conclusion, shouldn't the spread between the SparseCL scores in the Contradiction and Entailment be wide?  They seem to be no wider apart than the scores for the other methods.
> > * I can see that there is definitely some effect taking place.  On the other hand, doesn't the sparsity of the SparseCL embeddings cause the numbers in these rows to not be strictly comparable in any case?
> >
> > In Section 4.6 (the augmented Arguana ablation) seems to be reliant on the same synthetic data source.  So, again, there are methodological issues here (IMHO).
> >
> >
> > My rating remains unchanged.

---

> > > ### Author Response · Authors · 2025-04-07
> > >
> > > Thank you for your feedback.
> > >
> > > **Response regarding the adapted MSMARCO and HotPotQA datasets:**
> > >
> > > We will be happy to follow the reviewer's suggestion and use different names for these datasets throughout the paper. That said, given that we clearly explain how these datasets were constructed in the introduction, Section 4.2, Section 4.3, and elsewhere, we do not believe there is any confusion about how they were generated.
> > >
> > > In addition, we welcome any constructive suggestions for other datasets on which we could evaluate our method.
> > >
> > > **Response to more detailed issues raised:**
> > >
> > > In our “data set construction” paragraph in section 4.2, after generating the paraphrases $\{x^+_1,x^+_2,x^+_3\}$ and contradictions $\{x^-_1,x^-_2,x^-_3\}$ for the original document $x$, we remove $x$ from the corpus. After that, only the generated content remains in the corpus for use in subsequent queries and query answers. This ensures that all documents used in our experiments share the same writing style—generated by the same language model, with the same style control instruction included in both prompts. Therefore, we believe there is no detectable “LLM indicator” that exists only within a subset of our corpus, aside from the natural stylistic differences between paraphrases and contradictions.
> > >
> > > Regarding the inherent language style difference between a human-written text and a text written to contradict it, we conducted an experiment to verify that the word “not” appears more frequently in human-written contradictions. Please see the statistics below for the Arguana corpus.
> > >
> > > We counted the number of times “not” appears at the beginning of each argument or counter-argument in the Arguana corpus:
> > >
> > > Among the query arguments: “not” appears 614 times
> > >
> > > Among the counter-arguments: “not” appears 1,274 times
> > >
> > >
> > > We can see that even in human-written contradiction pairs, certain inherent language style differences are inevitable. Given that existing baseline methods do not perform well on Arguana even after fine-tuning, we believe that this style difference is an inherent aspect of contradiction retrieval tasks, rather than a flaw.
> > >
> > > **Response regarding our NLI experiments:**
> > >
> > > The goal of our method is to derive a score function that assigns the highest scores to contradiction pairs (as illustrated in Figure 2), and lower scores to other types of relations (such as paraphrases and random pairs). Based on this score function, the retrieval algorithm then searches for the maximal score within the dataset to identify contradictions for a given query.
> > >
> > > That said, numerical comparisons across different rows are not the focus; what matters is whether the highest score within each row is assigned to the contradiction pair, and whether the gap between contradictions and the other two classes is sufficiently large.
> > >
> > > From Table 5 and our analysis in Section 4.5, we observe that—qualitatively—only our method consistently assigns the highest scores to contradictions while maintaining a non-negligible margin over the other two classes. This is the reason we bolded the statistics for our method. In contrast, the other two methods either failed to distinguish between contradictions and paraphrases, or assigned the highest scores to paraphrases instead of contradictions.

---

### Official Review · Reviewer_RTuC · 2025-03-25

**Overall Recommendation:** 4

**Summary:**

* Introduces a contradiction retrieval method using cosine similarity and Hoyer measure of sparsity. The novel idea being using sparsity of embedding differences as a function to model contradiction.
* Discusses other approaches namely bi-encoders and cross-encoder models along with their shortcomings and addresses how SparseCL overcomes those.
* Shows robustness of the proposed fine tuning method on three pretrained sentence embedding models of different sizes across three datasets: Arguana, adapted HotpotQA and MSMARCO.
* Discusses imperfections of Arguana dataset, augments with paraphrases to make it harder and shows differential performance of methods as the number of paraphrases increase.
* Shows generalization capability of the sparse-aware embeddings by training on HotpotQA and testing on MSMARCO and, vice versa.
* Applies the method on two downstream applications of corpus cleaning and NLI.

**Claims And Evidence:**

* Claims that transitivity of cosine function makes it inherently incapable of representing the contradiction relation, backs it up with proofs and shows an average improvement of 3.6% with sparsity aware embeddings.
* The method enhances the speed of contradiction detection, supported by an experiment showing 200 times speed up compared to cross encoder methods.
* Mentions “embeddings designed to preserve subtle, contradictory nuances between sentences” and “effectively captures the essence of contradiction”, but lacks qualitative evidence to support these claims.

**Essential References Not Discussed:**

NA

**Experimental Designs Or Analyses:**

Checked the soundness of Counter-argument Retrieval, Contradiction retrieval, Zero-shot Generalization Tests and Arguana retrieval results analysis.

Very thorough job explaining details except in Contradiction retrieval task, It is not clear whether both, Zeroshot(cosine) and CL(cosine) are expected to give high scores for similar vs contradictory examples (seemingly zeroshot/pre-trained embeddings are trained on semantic similarity whereas CL uses contradictions as positives), and if either scenario impacts the validity of comparing these two methods.

**Methods And Evaluation Criteria:**

* Cosine similarity being limiting and using spasticity of embedding difference on top, is a good idea to retrieve contradictions. Although it is not very clear if Hoyer Sparsity is able to model and distinguish between contradictory differences vs orthogonal differences between two embeddings.
* Breadth of benchmarking datasets and data augmentations shows effectiveness of the approach.
* For the evaluation metric, most of the earlier work used Recall@1 and the paper uses NDCG@10, explanation for the choice of hyperparameter would make the shift clearer.

**Other Comments Or Suggestions:**

* [Typo] Figure 2: sub figures are referred to as left and right, instead of top and bottom.
* Scoring function for contradiction retrieval: In the Problem Formulation, query passage is referred to as q and corpus passages with p1, p2 and so on. The score function should probably be calculated between q and corpus passages, pi.

**Other Strengths And Weaknesses:**

* A novel idea to use sparsity of differences between the sentence embeddings to model contradiction.
* Well written, provides theoretical and quantitative backing(supplement section C and G) on claims against the existing bi-encoder and cross-encoder models.
* Shows robustness of the proposed method on three embedding models of different sizes and across three datasets.
* An insightful ablation into the limitations of Arguana dataset (section 4.6), proposed augmentations and effectiveness of sparsity-based retrieval over similarity based methods(supplement section E).
* Some arguments lack evidence/clarity e.g. “embeddings designed to preserve subtle, contradictory nuances between sentences”, meaning of the scores from Zero Shot (Cosine) vs CL (Cosine) methods.

**Questions For Authors:**

NA

**Relation To Broader Scientific Literature:**

* Introduces a method for contradiction retrieval task, using cosine similarity and Hoyer measure of sparsity, presenting sparsity of embedding differences as a function to capture contradiction, being the novel idea.
* Discusses prior similarity based methods for contradiction retrieval like bi encoders and cross encoders and their limitations.
* Runs two small experiments to show their method for contradiction retrieval can be applied to downstream applications of corpus cleaning and NLI.
* Talks about how contradiction retrieval is related to some major LLM based research efforts in recent years namely Fact verification and LLM hallucination, and augmented LLM and retrieval corpus attack.

**Theoretical Claims:**

NA

---

> ### Author Rebuttal · Authors · 2025-04-01
>
> We greatly appreciate your insightful comments on our work!
>
> Thank you for pointing out the typos.
>
> **"Some arguments lack evidence/clarity e.g. “embeddings designed to preserve subtle, contradictory nuances between sentences”, meaning of the scores from Zero Shot (Cosine) vs CL (Cosine) methods."**
>
> This sentence serves as a motivating hypothesis that explains why we believe embeddings for contradictions exhibit sparse differences. We acknowledge the current lack of evidence in explaining the semantic meaning behind each coordinate, which is an intriguing area for further investigation.
>
> We will adjust the figure captions and the notations in the score function as you suggested.

---

### Official Review · Reviewer_V6ZT · 2025-04-06

**Overall Recommendation:** 4

**Summary:**

The paper is about non-similarity based information retrieval which is currently under explored. In the paper they introduce SparseCL, a novel approach to address shortcomings in existing similarity search and cross-encoder models when retrieving arguments contradictory to the query from large document corpora.  The proposed method utilizes a combined metric of cosine similarity and sparsity function to identify and retrieve documents that contradict a given query.

The proposed method shows an 11.0% improvement when evaluated on existing retrieval benchmarks.

In summary;
- The paper introduces a novel approach using sentence embeddings together with cosine similarity and Hoyer measure of sparsity to capture the essence of contradiction.
- The embedding and scoring proposed approach shows an improved performance as compared to existing methods.

**Claims And Evidence:**

The authors make a compelling argument that supports the proposed approach of training a sentence embedding model to preserve sparsity of differences between the contradicted sentence embeddings. This is represented in Figure 2. Hoyer sparsity histogram.

**Essential References Not Discussed:**

NA

**Experimental Designs Or Analyses:**

The experimental design is reasonable, the authors test the performance of the proposed method on well known benchmarks of Arguana (Wachsmuth et al., 2018) a counterargument retrieval task and two contradiction retrieval datasets adapted from HotpotQA (Yang et al., 2018) and MSMARCO (Nguyen et al., 2016).
They apply the proposed contradiction retrieval to downstream applications ie., retrieval corpus cleaning and natural language inference. In addition authors perform ablation to explain the functionality of each component of the proposed method.

**Methods And Evaluation Criteria:**

The proposed method is well supported by evaluations on well known benchmarks of Arguana (Wachsmuth et al., 2018) a counterargument retrieval task and two contradiction retrieval datasets adapted from HotpotQA (Yang et al., 2018) and MSMARCO (Nguyen et al., 2016).

**Other Comments Or Suggestions:**

NA

**Other Strengths And Weaknesses:**

The study makes an in-depth analysis of non-similarity based information retrieval, the analysis suggests that a combined metric of cosine similarity and sparsity function to identify and retrieve documents that contradict a given query improves performance as compared to existing methods.

The authors cite and explain the existing studies, they also make a good literature study of the existing and related studies and how their proposed approach addresses the similarity problem differently.

The paper is well written, with clear figures, and the experimental results support the findings.

**Questions For Authors:**

NA

**Relation To Broader Scientific Literature:**

Similarity retrieval is a well studied task where contrastive learning is used to map similar sentences together and dissimilar ones far from each other in the embedding space (Gao et al., 2021, Karpukhin et al., 2020 and Xiong et al., 2020). In the same spirit, the authors try to address a rather different problem of non-similarity based retrieval.

**Theoretical Claims:**

NA

---

### Decision · Program_Chairs · 2025-05-01

**Decision:**

Accept (poster)

**Comment:**

This paper introduces a novel approach to contradiction retrieval called SparseCL. The approach combines standard cosine similarity metrics with a sparsity function that allows for efficient and effective retrieval of documents that refute the given query. The approach is evaluated on both human-curated and synthetically generated data sets, showing consistent improvements over baselines.

The reviewers found the proposed approach to be novel and well motivated, the empirical results to be strong, the method to demonstrate convincing efficiency properties, and the presentation to be clear.

The reviewers also noted a number of weaknesses, the primary one being the reliance on synthetic data for some of the evaluations, and the fact that the approach did not work quite as well on manually curated data.

Overall, though, the strengths of the paper outweighed the weaknesses. Efficient contradiction retrieval is a practically important direction and an interesting research direction. This paper represents a meaningful contribution and therefore should be accepted for publication.